# Microsecond interaural time difference discrimination restored by cochlear implants after neonatal deafness

**Nicole Rosskothen-Kuhl[1,2][†]\*, Alexa N Buck[1][†], Kongyan Li[1], Jan WH Schnupp[1,3]\***

[1]Department of Biomedical Sciences, City University of Hong Kong, Hong Kong, China; [2]Neurobiological Research Laboratory, Section for Clinical and Experimental Otology, University Medical Center Freiburg, Freiburg, Germany; [3]CityU Shenzhen Research Institute, Shenzhen, China

**Abstract** Spatial hearing in cochlear implant (CI) patients remains a major challenge, with many early deaf users reported to have no measurable sensitivity to interaural time differences (ITDs). Deprivation of binaural experience during an early critical period is often hypothesized to be the cause of this shortcoming. However, we show that neonatally deafened (ND) rats provided with precisely synchronized CI stimulation in adulthood can be trained to lateralize ITDs with essentially normal behavioral thresholds near 50 μs. Furthermore, comparable ND rats show high physiological sensitivity to ITDs immediately after binaural implantation in adulthood. Our result that ND-CI rats achieved very good behavioral ITD thresholds, while prelingually deaf human CI patients often fail to develop a useful sensitivity to ITD raises urgent questions concerning the possibility that shortcomings in technology or treatment, rather than missing input during early development, may be behind the usually poor binaural outcomes for current CI patients.

**\*For correspondence:**
nicole.rosskothen-kuhl@uniklinik-freiburg.de (NR-K);
jan.schnupp@googlemail.com (JWHS)

[†]These authors contributed equally to this work

**Competing interests:** The authors declare that no competing interests exist.

## Introduction

For patients with severe to profound sensorineural hearing loss, cochlear implants (CIs) can be enormously beneficial, as they often permit spoken language acquisition, particularly when CI implantation takes place early in life (*Kral and Sharma, 2012*). Nevertheless, the auditory performance achieved by CI users remains variable and falls a long way short of natural hearing.

For example, good speech understanding in the presence of competing sound sources requires the ability to separate speech from background. This is aided by 'spatial release from masking', a binaural phenomenon, which relies on the brain's ability to process binaural spatial cues, including interaural level and time differences (ILDs and ITDs) (*Ellinger et al., 2017*). While bilateral cochlear implantation is becoming more common (*Litovsky, 2010*; *Conti-Ramsden et al., 2012*; *Ehlers et al., 2017*), bilateral CI recipients still perform poorly in binaural tasks such as sound localization and auditory scene analysis, particularly when multiple sound sources are present (*van Hoesel, 2004*; *van Hoesel, 2012*). Indeed, while normal hearing (NH) human listeners may be able to detect ITDs as small as 10 μs (*Zwislocki and Feldman, 1956*), ITD sensitivity of CI patients, particularly with prelingual onset of deafness, is often poor and sometimes seems completely absent (*van Hoesel, 2004*; *Litovsky, 2010*; *van Hoesel, 2012*; *Kerber and Seeber, 2012*; *Litovsky et al., 2012*; *Laback et al., 2015*; *Ehlers et al., 2017*).

The reasons for the poor binaural sensitivity of CI recipients are only poorly understood, but two main factors are generally thought to be chiefly responsible, namely: (1) technical limitations of current CI devices and (2) neurobiological factors, such as when the neural circuitry responsible for processing binaural cues fails to develop due to a lack of experience during a presumed 'critical period' in early life or when it degenerates during a period of late deafness. These presumed factors

could act alone or in combination. Let us first consider the technological issues. The vast majority of CI devices in clinical use operate stimulation, which are variants of the 'continuous interleaved sampling' (CIS) method (*Wilson et al., 1991*). While these technical limitations are substantial, currently few researchers believe that they alone can be fully responsible for the poor binaural acuity observed in CI patients because it is possible to test patients with experimental processors that overcome some of the shortcomings of standard issue clinical devices. When tested with such experimental devices, many postlingually deaf CI users show better ITD sensitivity, with some of the best performers achieving thresholds comparable to those seen in NH peers. In contrast, the ITD performance of prelingually deaf CI users remains invariably poor, with even rare star performers only achieving thresholds of a few hundred microsecond (*Poon et al., 2009*; *Conti-Ramsden et al., 2012*; *Litovsky et al., 2012*; *Gordon et al., 2014*; *Laback et al., 2015*; *Litovsky and Gordon, 2016*; *Ehlers et al., 2017*). It is this poor performance of prelingually deaf patients even under optimized experimental conditions that led to the suggestion that the absence of binaural inputs during a presumed 'critical' period in early childhood may prevent the development of ITD sensitivity (*Kral and Sharma, 2012*; *Kral, 2013*; *Litovsky and Gordon, 2016*; *Yusuf et al., 2017*).

In this context, it is however important to remember that the terms 'sensitive' and 'critical' period do not have simple, universally accepted definitions, which may create uncertainty about what exactly a 'critical period hypothesis of binaural hearing' proposes. Some authors distinguish 'strong' and 'weak' critical periods. Both types of critical periods are developmental periods during which the acquisition of a new sensory or sensory-motor faculty appears to be particularly easy. However, after 'weak' critical periods, a full mastery of a faculty may still be acquired with a little more effort (*Kilgard and Merzenich, 1998*), but missing essential experience during a 'strong' critical period leads to substantial and irreparable limitations later in life (*Knudsen et al., 1984*). Perhaps the best studied example of a strong critical period disorder is amblyopia. Amblyopic patients experience an uneven or unbalanced binocular visual stimulation in early life, which leads to a failure of the normal development of the brain's binocular circuitry. This, in turn, causes sometimes dramatic impairments in the visual acuity in the 'weaker eye', as well as in stereoscopic vision. These impairments can only be fully reversed if diagnosed and treated prior to critical period closure, and despite substantial research efforts, no interventions performed after critical period closure can offer more than partial remediation of the deficits (*Tsirlin et al., 2015*). If we hypothesize that binaural hearing development exhibits a similarly strong critical period, then developing clinical CI processors with better ITD coding might not benefit patients with hearing loss early in life, as it might not be possible to implant these patients early enough to provide them with suitable binaural experience during their (strong) critical period. Their brains would then be unable to learn to take full advantage of the binaural cues that improved CIs provided later in life might deliver.

For neonatally deaf patients, periods of sensory deprivation during development are the norm because profound bilateral hearing loss is hard to diagnose in neonates and measurements of auditory brainstem responses (ABRs) have to be repeated to exclude delayed maturation of the auditory brainstem (*Jöhr et al., 2008*; *Cosetti and Roland, 2010*; *Arndt et al., 2014*). Also, before CI surgery is considered non-invasive alternatives such as hearing aids may be tried first. Finally, risks associated with anesthesia in young babies provide another disincentive for very early implantation (*Dettman et al., 2007*; *Jöhr et al., 2008*; *Cosetti and Roland, 2010*). Altogether, this means by the time of implantation, neonatally deaf pediatric CI patients will typically already have missed out on many months of the auditory input. Consequently, if there is a strong binaural critical period, then this lack of early experience might put near-normal binaural hearing performance forever out of their reach.

Various lines of animal experimentation make such a critical period hypothesis plausible, including immunohistochemical studies that have shown degraded tonotopic organization (*Rosskothen-Kuhl and Illing, 2012*; *Rauch et al., 2016*) and changes in stimulation-induced molecular, cellular, and morphological properties of the auditory pathway of neonatally deafened (ND) CI rats (*Illing and Rosskothen-Kuhl, 2012*; *Rosskothen-Kuhl and Illing, 2012*; *Jakob et al., 2015*; *Rauch et al., 2016*; *Rosskothen-Kuhl et al., 2018*). Additional studies demonstrate that abnormal sensory input during early development can alter ITD tuning curves in key brainstem nuclei of gerbils (*Seidl and Grothe, 2005*; *Beiderbeck et al., 2018*). Furthermore, numerous electrophysiological studies on cats and rabbits have reported significantly lower ITD sensitivity to CI stimulation in the inferior colliculus (IC) (*Hancock et al., 2010*; *Hancock et al., 2012*; *Hancock et al., 2013*;

*Chung et al., 2019*) and auditory cortex (AC) (*Tillein et al., 2010*; *Tillein et al., 2016*) after early deafening compared to what is observed in hearing experienced controls.

However, although the 'strong critical period hypothesis' of poor ITD sensitivity in CI patients is plausible, it has not yet been rigorously tested. The previous animal studies just mentioned have not investigated perceptual limits of binaural function in behavioral experiments using optimized binaural inputs. Similarly, while we do know that CI patients with NH experience in early childhood usually have a better ITD sensitivity than patients without (*Litovsky, 2010*; *Laback et al., 2015*; *Ehlers et al., 2017*), we do not yet know whether early deaf patients could develop good ITD sensitivity after implantation later in life if they were fitted with CI processors providing with optimized binaural stimulation from the outset. Currently, only research interfaces that are unsuitable for everyday clinical use can deliver the high-quality binaural inputs needed to investigate this question. Consequently, there are currently no patient cohorts who experienced through their CIs the long periods of high-quality ITD information that may be needed for them to become expert at using ITDs, irrespective of any hypothetical critical periods. We cannot at present exclude the possibility that the ND auditory pathway may retain a substantial innate ability to encode ITD even after long periods of neonatal deafness, but that this ability may atrophy after countless hours of binaural CI stimulation through conventional clinical processors which convey no useful ITD information.

Since these possibilities cannot currently be distinguished based on clinical data, animal experimentation is needed, which can measure binaural acuity behaviorally. The first objective here is to examine how much functional ITD sensitivity can be achieved in mature ND animals, which receive bilaterally synchronized CI stimulation capable of delivering ITD cues with microsecond accuracy. Achieving this first objective was the aim of this article. In essence, we attempted to disprove the 'strong critical period hypothesis for ITD sensitivity development' by examining whether experimental animals fitted with binaural CIs may be able to achieve good ITD sensitivity without excessive training or complicated interventions, even after severe hearing loss throughout infancy. To achieve this, we used a stimulation optimized for ITD encoding straight after implantation.

We therefore established a new behavioral bilateral CI animal model and setup capable of delivering microsecond precise ITD cues to cohorts of ND rats (early-onset deafness), which received training with synchronized bilateral CI stimulation in young adulthood. These young adult rats learned easily and quickly to lateralize ITDs behaviorally, achieving thresholds as low as ~50 µs, comparable to those of their NH litter mates. We also observed that such ND rats exhibit a great deal of physiological ITD sensitivity in their IC straight after implantation. Our results therefore indicate that, at least in rats, there appears to be no strong critical period for ITD sensitivity.

## Results

### Early deaf CI rats discriminate ITD as accurately as their normally hearing litter mates

To test whether ND rats can learn to discriminate ITDs of CI stimuli, we trained five ND rats that received chronic bilateral CIs in young adulthood (postnatal weeks 10–14) in a two-alternative forced choice (2AFC) ITD lateralization task (NDCI-B; see *Figure 1*), and we compared their performance against behavioral data from five age-matched NH rats trained to discriminate the ITDs of acoustic pulse trains (NH-B; see *Figure 1*; *Li et al., 2019*). Animals initiated trials by licking a center 'start spout' and responded to 200 ms long 50 Hz binaural pulse trains by licking either a left or a right 'response spout' to receive drinking water as positive reinforcement (*Figure 2—figure supplements 1a* and *2b*; *Video 1*). Which response spout would give water was indicated by the ITD of the stimulus. We used pulses of identical amplitude in each ear, so that systematic ITD differences were the only reliable cue available (*Figure 2—figure supplements 1c,d* and *2f*). NDCI-B rats were stimulated with biphasic electrical pulse trains delivered through chronic CIs, NH-B rats received acoustic pulse trains through a pair of 'open stereo headphones' implemented as near-field sound tubes positioned next to each ear when the animal was at the start spout (*Figure 2—figure supplements 2a*, see *Li et al., 2019* for details). During testing, stimulus ITDs varied randomly.

The behavioral data (*Figure 2*) were collected over a testing period of around 14 days. For NDCI-B rats, the initial lateralization training started usually 1 day after CI implantation. On average, rats were trained for 8 days before we started to test them on ITD sensitivity. The behavioral

**Figure 1.** Timeline and experimental treatment of our three cohorts. NDCI-B and NDCI-E rats were both neonatally deafened by kanamycin and bilaterally implanted as young adults. Around half of them went into a behavioral training and testing (NDCI-B), while the other half were used for multi-unit recordings of IC neurons directly after bilateral CI implantation. NH-B rats were normal hearing and started a behavioral training and testing as young adults. w: weeks. d: days.

performance of each rat is shown in *Figure 2*, using light blue for NH-B (a–e) and dark blue for NDCI-B (f–j) animals. *Figure 2* clearly demonstrates that all rats, whether NH with acoustic stimulation or ND with CI stimulation, were capable of lateralizing ITDs. As might be expected, the behavioral sensitivity varied from animal to animal. To quantify the behavioral ITD sensitivity of each rat, we fitted psychometric curves (see Materials and methods, red lines in *Figure 2*) to the raw data and calculated the slope of that curve at ITD = 0. *Figure 2k* summarizes these slopes for NH-B (light blue) and NDCI-B (dark blue) animals.

The slopes for both groups fell within the same range. Remarkably, the observed mean sensitivity for the NDCI-B animals (0.487%/μs) is only about 20% worse than that of the NH-B (0.657%/μs). Furthermore, the differences in means between experimental groups (0.17%/μs) were so much smaller than the animal-to-animal

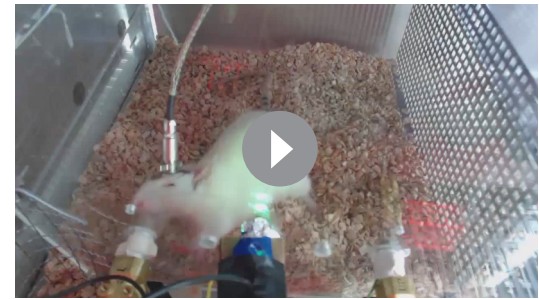

**Video 1.** Neonatally deafened CI rat performing a two-alternative forced choice ITD lateralization task in a custom-made behavior setup. The animal initiates trials by licking the center 'start spout' and responds to binaural pulse trains by licking either the left or right 'response spout' to receive drinking water as positive reinforcement if the response is correct or a time out with the flashing light as negative reinforcement if the response is incorrect. Which response was correct was indicated by the ITD stimulus presented on that trial when the animal licks the center spout.
https://elifesciences.org/articles/59300#video1

variance (~0.73 %$^2$/μs$^2$) that prohibitively large cohorts of animals would be required to have any reasonable prospect of finding a significant difference. Indeed, we performed a Wilcoxon test on the slopes and found no significant difference (p=0.4375). Similarly, both cohorts showed similar 75% correct lateralization thresholds (median NH-B: 41.5 μs; median NDCI-B: 54.8 μs; mean NH-B: 79.9 μs; mean NDCI-B: 63.5 μs). Remarkably, the ITD thresholds of our ND-CI rats are thus orders of magnitude better than those reported for prelingually deaf human CI patients, who often have ITD thresholds too large to measure and often in excess of 3000 μs (*Litovsky et al., 2010*; *Ehlers et al., 2017*). Thresholds in ND rats were not dissimilar from the approx. 10–60 μs range of 75% correct ITD discrimination thresholds reported for normal human subjects tested with noise bursts (*Klumpp and Eady, 1956*), and pure tones (*Zwislocki and Feldman, 1956*), or the ≈40 μs thresholds reported for NH ferrets tested with noise bursts (*Keating et al., 2013a*).

## Varying degrees and types of ITD tuning are pervasive in the neural responses in the IC of ND rats immediately after adult cochlear implantation

To investigate the amount of physiological ITD sensitivity present in the hearing inexperienced rat brain, we recorded responses of n = 1140 multi-units in the IC of four young adult ND rats (NDCI-E; see *Figures 1* and *3*). These rats were litter mates of the behavioral ND animals (NDCI-B) and were stimulated by isolated, bilateral CI pulses with ITDs varying randomly over a ±160 μs range (ca 123% of the rat's physiological range *Koka et al., 2008*). For the cohort of neonatally deafened rats (NDCI-E), the CI implantation and the electrophysiology measurements in the IC were performed on the same day with no prior electric hearing experience. The stimuli were again biphasic current pulses of identical amplitude in each ear, so that systematic differences in responses can only be attributed to ITD sensitivity (see *Figure 2—figure supplements 1c-d*). Responses of IC neurons were detected for currents as low as 100 μA. *Figure 3* shows a selection of responses as raster plots and corresponding ITD tuning curves, as a function of firing rate (*Figure 3*, #1–4).

For NDCI-E animals, ITD tuning varied from one recording site to the next both in shape and magnitude and firing rates clearly varied as a function of ITD values (*Figure 3*). While many multi-units showed typical short-latency onset responses to the stimulus with varying response amplitudes (*Figure 3*, #1, #3), some showed sustained, but still clearly tuned, responses extending for up to 80 ms or longer post-stimulus (*Figure 3*, #4). Although the interpretation of tuning curves is complex the shapes of ITD tuning curves we observed in rat IC (*Figure 3*) resembled mostly the 'peak', 'monotonic sigmoid', 'trough', and 'multi-peak' shapes previously described in the IC of cats (*Smith and Delgutte, 2007*).

To quantify how strongly the neural responses recorded at any one site depended on stimulus ITD, signal-to-total variance ratio (STVR) values were calculated as described in *Hancock et al., 2010*. It quantifies the proportion of response variance that can be accounted for by stimulus ITD (see Materials and methods). Each sub-panel of *Figure 3* indicates STVR values obtained for the corresponding multi-unit, while *Figure 4* shows the distributions of STVR values for the NDCI-E cohort (red). For comparison with a similar previous bilateral CI study, *Figure 4* also shows the STVR values for the IC of congenitally deaf (blue) cats reported by *Hancock et al., 2010* and in which they are referred to as signal-to-noise (SNR) values. When comparing the distributions shown, it is important to be aware that there are significant methodological, as well as species, differences between our study and the study that produced the cat data shown in *Hancock et al., 2010*, so the cross-species comparison in particular must be done with care. Nevertheless, the distributions clearly show that ITD STVRs in our NDCI-E rats are very good, and also in line with the values reported by others using similar methodologies. For interpretation purposes, an STVR > 0.5 is considered good ITD sensitivity. It is noticeable that the proportion of multi-units with relatively large STVRs values (substantial ITD tuning) is high among the NDCI-E rats with a median STVR value for IC multi-units of 0.362. In comparison, *Hancock et al., 2010* showed a lower median STVR (referred to as SNR) for congenitally deaf cats (0.19) as compared to adult deafened cats (0.45). The proportion of rat multi-units that showed statistically significant ITD tuning (p≤0.01), as determined by ANOVA (see Materials and methods), was also very large in ND (1125/1229 ≈ 91%) CI-stimulated rats. Thus, for our rats that were deafened before the onset of hearing, a lack of early auditory experience during what ought to have been a critical period for ITD sensitivity did not produce a measurable decline in overall sensitivity of IC neurons to the ITD of CI stimulus pulses. This is perhaps unexpected given

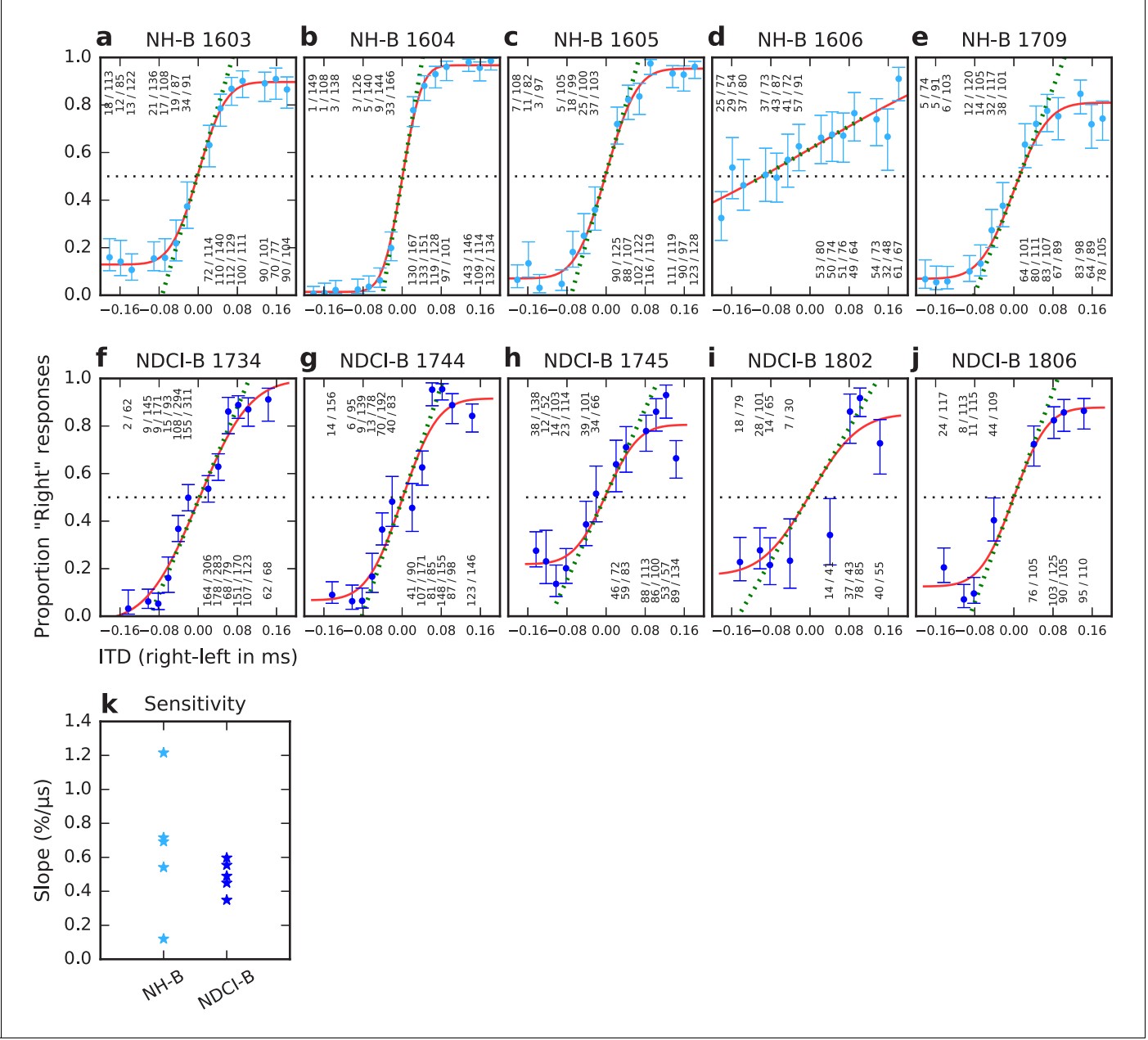

**Figure 2.** ITD psychometric curves of normal hearing acoustically stimulated (NH-B, **a–e**) and neonatally deafened CI-stimulated rats (NDCI-B, **f–j**). Panel titles show corresponding animal IDs. Y-axis: proportion of responses to the right-hand side. X-axis: Stimulus ITD in ms, with negative values indicating left ear leading. Blue dots: observed proportions of 'right' responses for the stimulus ITD given by the x-coordinate. Number fractions shown above or below each dot indicate the absolute number of trials and 'right' responses for corresponding ITDs. Blue error bars show Wilson score 95% confidence intervals for the underlying proportion 'right' judgments. Red lines show sigmoid psychometric curves fitted to the blue data using maximum likelihood. Green dashed lines show slopes of psychometric curves at x = 0. Slopes serve to quantify the behavioral sensitivity of the animal to ITD. Panel (**k**) summarizes the ITD sensitivities (psychometric slopes) across the individual animal data shown in (**a–j**) in units of % change in animals' 'right' judgments per µs change in ITD.

The online version of this article includes the following figure supplement(s) for figure 2:

**Figure supplement 1.** Bilateral electrical intracochlear stimulation of cochlear implanted (CI) rats.

**Figure supplement 2.** Bilateral psychoacoustics near-field setup for normal hearing (NH) rats.

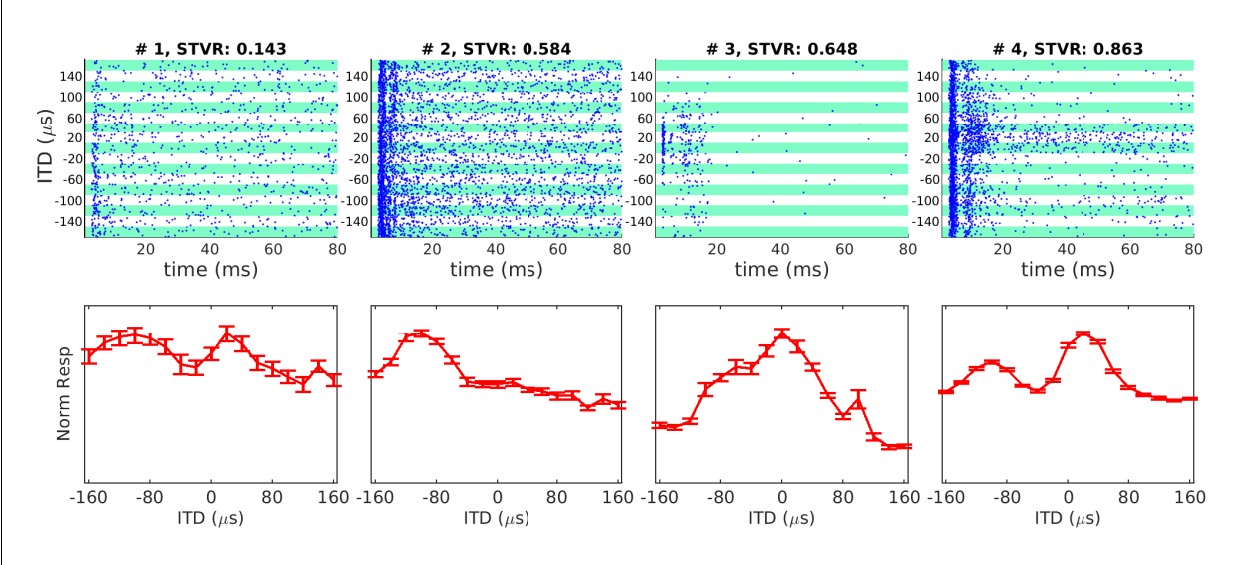

**Figure 3.** Examples of interaural time difference (ITD) tuning curves of neonatally deafened CI rats (NDCI-E) as a function of ITD. Dot raster plots are shown above the corresponding ITD tuning curves. The multi-units shown were selected to illustrate some of the variety of ITD tuning curve depths and shapes observed. In the raster plots, each blue dot shows one spike. Alternating white and green bands show responses to n = 30 repeats at each ITD shown. Tuning curve response amplitudes are baseline corrected and normalized relative to the maximum of the mean response across all trials, during a period of 3–80 ms post-stimulus onset. Error bars show SEM. Above each sub-panel we show signal-to-total variance ratio (STVR) values to quantify ITD tuning. Panels are arranged top to bottom by increasing STVR. ITD > 0: ipsilateral ear leading; ITD < 0: contralateral ear leading.

that previous studies comparing ITD tuning in the IC of congenitally deaf white cats with that of hearing experienced, adult deafened wild type cats did report noticeably worse ITD tuning in the congenitally deaf cats (*Hancock et al., 2013*). Note that congenitally deaf white cats lose hair-cell function between postnatal days 3 and 10 (*Mair and Elverland, 1977*).

Nevertheless, the results in *Figures 3* and *4* clearly show that many IC neurons in the inexperienced, adult midbrain of NDCI-E rats are quite sensitive to changes in ITD of CI pulse stimuli by just a few tens of microsecond, and our behavioral experiments showed that NDCI-B rats (*Figure 2f–j*) can readily learn to use this neural sensitivity to perform behavioral ITD discrimination with an accuracy similar to that seen in their NH-B litter mates (*Figure 2a–e*).

## Discussion

This study is the first demonstration that, at least in rats, severely degraded auditory experience in early development does not inevitably lead to impaired binaural time processing in adulthood. In fact, the ITD thresholds of our NDCI-B rats (≈ 50 μs) were as good as the ITD thresholds of NH-B rats *Li et al., 2019*, and many times better than those typically reported for early deaf human CI patients with thresholds often too large to measure (*Litovsky et al., 2010*; *Gordon et al., 2014*; *Ehlers et al., 2017*). The good performance exhibited by our NDCI-B animals raises the important question of whether early deaf human CI patients might perhaps also be able to achieve near-normal ITD sensitivity if supplied with optimal bilateral CI stimulation capable of delivering adequate ITDs from the first electric stimulation even in the absence of hearing experience during a period what has been thought to be critical for the development of ITD sensitivity. But before we consider translational questions that might be raised by our results, we should address two aspects of this study, which colleagues may find surprising:

First, some studies deemed rats to be a poor model for ITD processing due to medial superior olives (MSOs) with less ITD-sensitive neurons and their limited low-frequency hearing that may result in limited ITD perception (*Grothe and Klump, 2000*; *Wesolek et al., 2010*). However, in animals with relatively high-frequency hearing, such as rats, envelope ITD coding through the lateral superior olive is likely to make important contributions (*Joris and Yin, 1995*). The only previous behavioral study of ITD sensitivity in rats outside of our lab (*Wesolek et al., 2010*) concluded that rats are

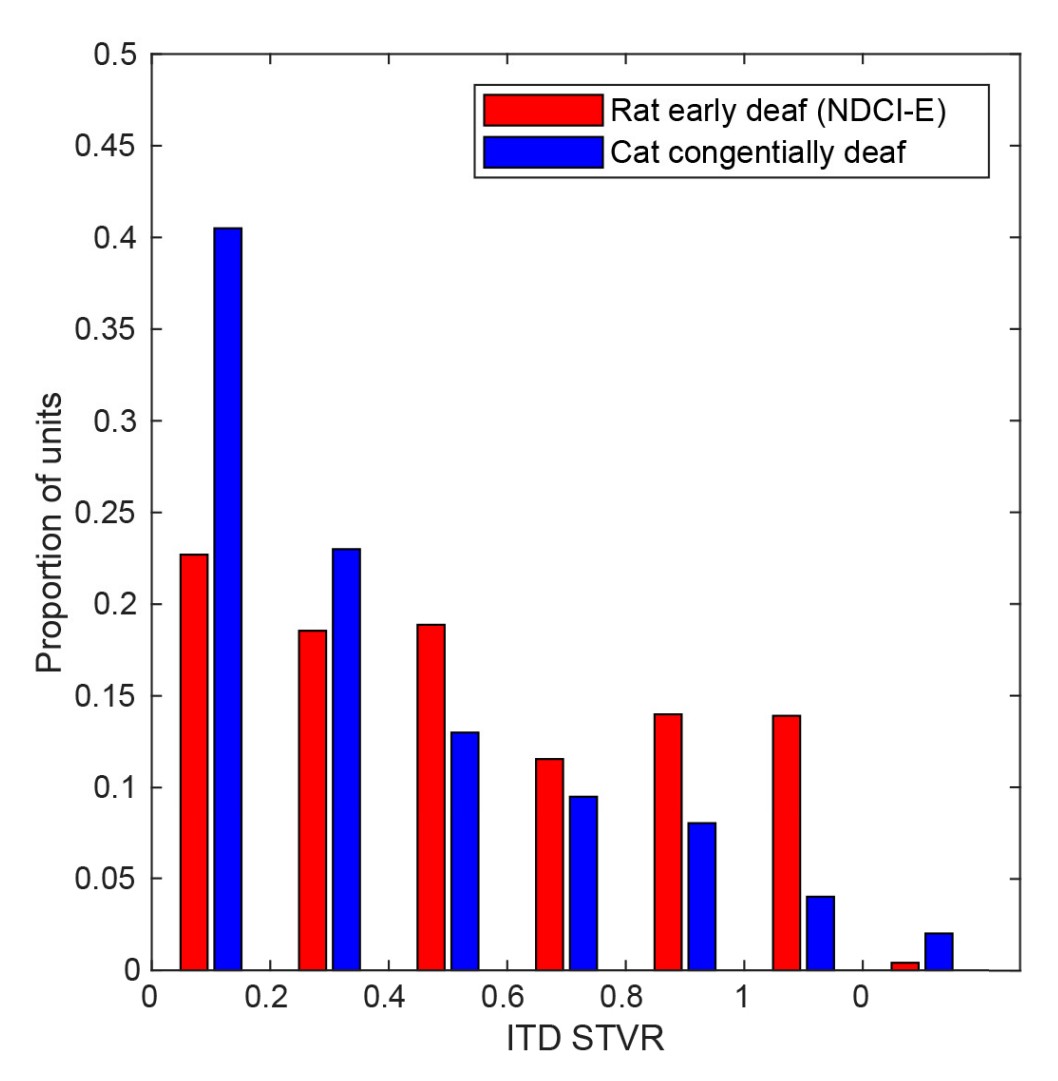

**Figure 4.** Bar chart shows distributions of interaural time difference (ITD) signal-to-total variance ratio (STVR) values for inferior colliculus (IC) multi-units of neonatally deafened cochlear implanted rats (NDCI-E, red). STVR value distributions for IC single-unit data recorded by *Hancock et al., 2010* for congenitally deaf cats (blue) under cochlear implant stimulation are also shown for comparison and are referred to as SNR in this cat study.

unlikely to be sensitive to the interaural phase of relatively low-frequency tones. High-frequency 'envelope' ITD sensitivity is also bound to be of great importance in prosthetic hearing given that CIs rarely reach the apex of the cochlea. In *Li et al., 2019*, we recently demonstrate that NH rats can use ITD cues for 2AFC sound lateralization tasks and thus conclude that, at least to broadband clicks, rats show ITD sensitivity. Here, we focused on broadband acoustic or electrical pulse stimuli that provide plenty of 'onset' and 'envelope' ITDs and that are processed well even at high carrier frequencies (*Joris and Yin, 1995*; *Bernstein, 2001*). That may also explain why our CI rats showed good ITD sensitivity even though our CIs targeted the lower mid-frequency region in each ear, and not the apical region associated with low-frequency hearing. Recent studies in CI patients with late deafness in adults and children have shown that ITDs delivered to mid- and high-frequency cochlear regions can be detected behaviorally (*Kan et al., 2016*; *Ehlers et al., 2017*).

Second, electrophysiology studies on congenitally deaf CI cats reported a substantially reduced ITD sensitivity relative to wild-type, hearing experienced cats acutely deafened as adults (*Tillein et al., 2010*; *Hancock et al., 2010*; *Tillein et al., 2016*). These studies recorded neural tuning high up in the auditory pathway (AC and IC); therefore, one cannot be certain whether the reduced

sensitivity reflects a fundamental degradation of ITD processing in the olivary nuclei or merely poor maturation of connections to higher order areas, the latter of which may be reversible with experience and training. In the IC of our ND rats, we found significant ITD sensitivity in 91% of recordings sites, compared to only 48% reported for congenitally deaf cats (*Hancock et al., 2010*) or 62% for ND rabbits (*Chung et al., 2019*). When tested under optimal stimulation to deliver microsecond precise ITD cues in a naive auditory system, the proportion of ITD-sensitive sites in NDCI-E rats is thus more similar to proportions in adult deafened CI-stimulated cats (84–86%; *Smith and Delgutte, 2007*; *Hancock et al., 2010*), rabbits (~75%; *Chung et al., 2016*; *Chung et al., 2019*), or gerbils (~74%; *Vollmer, 2018*). *Figure 4* suggests that the ITD STVR seen in our ND-CI rats fall in a similar range of the ITD STVRs previously reported for congenitally deaf CI cats (*Hancock et al., 2010*). While for cats, the proportion of IC multi-units with large ITD STVR values (>0.5) appears to be reduced in animals lacking early auditory experience, the same does not appear to be the case in our rats. Whether these quantitative differences in physiological ITD sensitivity are due to methodological and/or species differences is not determinable, but we believe that these apparent differences are ultimately unlikely to be important because even the congenitally deaf cats still have a decent number of IC units showing fairly large amounts of ITD sensitivity. In fact, more than 20% of the congenitally deaf cat IC units have STVR values of 0.5 or higher, which indicates rather good ITD sensitivity. It is important to remember that it is unknown how much ITD tuning in the IC or AC is necessary, or whether this is species specific, to make behavioral ITD discrimination thresholds of ≈ 50 μs possible, as we see in our NDCI-B cohort (*Figure 2*). However, multi-units such as #3 and #4 shown in *Figure 3* change their firing rates as a function of ITD substantially between steps of only 20 μs. These multi-units have STVRs that are not outside the range reported for congenital deaf cats. Thus we cannot conclude from the electrophysiology data alone whether the quantitative differences in ITD sensitivity between these studies would equate to difference in behavioral lateralization performances. The level of physiological ITD sensitivity previously observed in cats (*Tillein et al., 2010*; *Hancock et al., 2010*; *Tillein et al., 2016*) and rabbits (*Chung et al., 2019*) could be sufficient to enable good behavioral ITD discrimination performance if only these animals could be trained and tested on an appropriate task.

Thus, in our opinion, any previously reported reductions in physiological ITD sensitivity seen in the IC (*Chung et al., 2016*; *Chung et al., 2019*) or AC (*Tillein et al., 2010*; *Tillein et al., 2016*) of early deaf animals does not seem nearly large enough to fully explain the very poor behavioral ITD thresholds seen in most early deaf humans. And indeed, our own findings that the behavioral ITD sensitivity of our NDCI-B rats compares favorably with that in NH-B animals strongly suggests that the poor ITD sensitivity in human CI patients may well have causes beyond the lack of auditory experience during a presumed critical period.

It is unclear why congenitally deaf cats (*Hancock et al., 2010*; *Hancock et al., 2013*) show a modest reduction in neural ITD sensitivity, while NDCI-E rats in the present study seem not to. There are numerous methodological and species differences that might account for this, ultimately relatively small discrepancy. More pertinent for the present discussion is that both preparations exhibit a lot of innate residual ITD sensitivity in their midbrains despite severe hearing impairment throughout their development (*Tillein et al., 2010*; *Hancock et al., 2010*; *Hancock et al., 2013*; *Tillein et al., 2016*; *Chung et al., 2019*). We would find it surprising if this physiological ITD sensitivity of IC neurons could not ever be harnessed to inform perceptual decisions in the cats' and rabbits' brains. Thus, in light of our new behavioral results in rats, it seems reasonable to expect that, with appropriate rehabilitation and training, neonatally deaf cats and rabbits (and perhaps even humans) might be able to learn to make use of their residual innate physiological ITD sensitivity to perform very well in binaural hearing tasks.

The most striking difference between our results and other previously published work remains that the behavioral ITD discrimination thresholds of our NDCI-B rats are an order of magnitude or more better than those of early deaf human CI patients (*Gordon et al., 2014*; *Litovsky and Gordon, 2016*; *Ehlers et al., 2017*). As mentioned in the introduction, previous authors have proposed that the very poor performance typical of early deaf human CI patients may be due to 'factors such as auditory deprivation, in particular, lack of early exposure to consistent timing differences between the ears' (*Ehlers et al., 2017*), in other words, the critical period hypothesis. However, neonatal deafening and severe hearing loss until reaching developmental maturity did not prevent our NDCI-B rats from achieving very good ITD discrimination performance. Admittedly, there may be

species differences at play here. Our ND rats were implanted as young adults, and were severely deprived of auditory input throughout their childhood, but humans mature much more slowly, so even patients implanted at a very young age will have suffered auditory deprivation for a substantially longer absolute time period than our rats. Nevertheless, our results on early deafened CI rats show that lack of auditory input during early development does not need to result in poor ITD sensitivity and is therefore unlikely to be a sufficient explanation for the poor ITD sensitivity found in early deaf CI patients.

Previous studies of the development of the binaural circuitry in animal models also have not provided any strong evidence for a critical period, even if they have pointed to the important role that early experience can have in shaping this circuitry. Most of these studies have focused on low-frequency fine-structure ITD pathways through the MSO, rather than lateral superior olive (LSO) envelope ITD pathways that are likely to be of particular relevance for the CI ITD processing we are studying here. But even if they may not be directly applicable, they are nonetheless somewhat informative for the present discussion. For example, developmental studies in ferrets have shown that the formation of afferent synapses to MSO, one of the main brainstem nuclei for ITD processing, is essentially complete before the onset of hearing (*Brunso-Bechtold et al., 1992*). In mice, the highly specialized calyxes of Held synapses, which are thought to play key roles in relaying precisely timed information in the binaural circuitry, have been shown to mature before the onset of hearing (*Hoffpauir et al., 2006*). In both cases, crucial binaural circuitry elements are completed before any sensory input dependent plasticity can take place. However, there are also studies that do indicate that the developing binaural circuitry can respond to changes in input. For example, in gerbils, key parts of the binaural ITD processing circuitry in the auditory brainstem will fail to mature when driven with strong, uninformative omnidirectional white noise stimulation during development (*Kapfer et al., 2002*; *Seidl and Grothe, 2005*), which shows that inappropriate or uninformative sensory input can disturb the development of binaural brainstem pathways. A perhaps related finding by *Tirko and Ryugo, 2012* shows that inhibitory pathways in the MSO, which are thought to be essential for ITD encoding, are significantly reduced in congenitally deaf cats at postnatal day 90, compared to NH peers, but they can be fully restored with the advent of CI stimulation after only 3 months. Finally, *Pecka et al., 2008* demonstrated the importance of glycinergic inhibition and its timing in the MSO in controlling binaural excitation by fine tuning the delay between arrival from the two ears, which could allow ITD pathways to be 'tuned', possibly in an experience dependent manner. Overall, these studies point to varying extents of experience dependence of the developing binaural pathway, but none of them would suggest that the absence of stimulation early in life would necessarily prevent the restoration of effective binaural processing after the closure of some presumed critical period. None of the published articles we could find point to a biological mechanism for a critical period that could explain the loss of ITD sensitivity in early deaf CI users merely as a consequence of an absence of input in early life.

It is well known that the normal auditory system not only combines ITD information with ILD and monaural spectral cues to localize sounds in space, it also adapts strongly to changes in these cues, and can re-weight them depending on their reliability (*Keating et al., 2013b*; *Keating et al., 2015*; *Tillein et al., 2016*; *Kumpik et al., 2019*). Similarly, *Jones et al., 2011* demonstrated changes in ITD and ILD thresholds as head size and pinnae grow for up to 6 weeks postnatally in chinchillas. Again this highlights the importance of plasticity of binaural hearing during development. However, no studies have demonstrated that critical periods in the ITD pathways will irrevocably close if sensory input is simply absent, rather than altered. By using the rat model, which allowed us to study ITD sensitivity behaviorally, we were able to show conclusively that the ability to use ITD cues perceptually does not disappear permanently after hearing loss during a presumed critical period.

Given that our results cast doubts on the critical period hypothesis, it may be time to consider other likely causes for ITD insensitivity in CI patients. One possibility that we believe has not been given enough attention is that an innate ITD sensitivity could conceivably degrade over the course of prolonged exposure to the entirely inconsistent and uninformative ITDs delivered by current standard clinical CI processors. This possibility is consistent with the observations by *Zheng et al., 2015* and by *Litovsky and Gordon, 2016* who note that, even after binaural CI listening experience extending for >4 years or >6 years, respectively, the ITD sensitivity of bilateral CI users still lags well behind that of age-matched controls with NH. If the clinical processors supplied to these bilateral CI users do not supply high-quality ITD cues, then no amount of experience will make these patients

experts in the use of ITDs. Contrast this with our NDCI-B rats, which received only highly precise and informative ITDs right from the start with no additional auditory cues, and were able to lateralize ITDs as well as their NH litter mates after only 2 weeks of training. This is admittedly a somewhat unfair comparison. Clincal CI processors are, for good reason, designed first and foremost for the purpose of delivering all important speech formant information in real life settings, and optimizing ITD coding was not a priority in their original design. Nevertheless, our results raise the possibility that incorporating better ITD encoding in clinical processors might lead to better binaural outcomes for future generations of CI patients.

Current CI processors produce pulsatile stimulation based on fixed rate interleaved sampling (**Wilson et al., 1991**; **Stupak et al., 2018**), which is neither synchronized to stimulus fine structure nor synchronized between ears. Furthermore, at typical clinical stimulation rates (>900 pps), CI users are not sensitive to speech envelope ITDs, as envelope ITD sensitivity requires peak shaped envelopes (**Laback et al., 2004**; **Grantham et al., 2008**; **van Hoesel et al., 2009**; **Noel and Eddington, 2013**; **Laback et al., 2015**). Consequently, the carrier pulses are too fast, and the envelope shapes in everyday sounds are not peaked enough, so that speech processors only ever provide uninformative envelope ITDs to the children using them (**Laback et al., 2004**; **Grantham et al., 2008**; **Laback et al., 2015**). Perhaps brainstem circuits of children fitted with conventional bilateral CIs simply 'learn' to ignore the unhelpful ITDs that are contained in the inputs they receive. This would mean that these circuits are adaptive to uninformative ITDs. In contrast, precise ITD cues at low pulse rates were essentially the only form of useful auditory input that our NDCI-B rats experienced, and they quickly learned to use these precise ITD cues. Thus, our data raise the possibility that the mammalian auditory system may develop ITD sensitivity in the absence of early sensory input and that this sensitivity may then be either refined or lost, depending on how informative the binaural inputs turn out to be.

The inability of early deaf CI patients to use ITDs may thus be somewhat similar to conditions such as amblyopia or failures of stereoscopic depth vision development, pathologies that are caused more by unbalanced or inappropriate inputs than by a lack of sensory experience (**Levi et al., 2015**). For the visual system, it has been shown that orientation selective neuronal responses exist at eye-opening and thus are established without visual input (**Ko et al., 2013**). If this hypothesis is correct, then it may be possible to 'protect' ITD sensitivity in young bilateral CI users by exposing them to regular periods of precise ITD information from the beginning of binaural stimulation. Whether CI patients are able to recover normal ITD sensitivity much later if rehabilitated with useful ITDs for prolonged periods, or whether their ability to process microsecond ITDs atrophies irreversibly, is unknown and will require further investigation.

While these interpretations of our findings would lead us to argue that bilateral CI processing strategies may need to change to make microsecond ITD information available to CI patients, one must nevertheless acknowledge the difficulty in changing established CI processing strategies. The CIS paradigm (**Wilson et al., 1991**) from which most processor algorithms are derived times the stimulus pulses, so that only one electrode channel delivers a pulse at any one time. This has been shown to minimize cross-channel interactions due to 'current spread' which might compromise the already quite limited tonotopic place coding of CIs. Additionally, CI processors run at high pulse rates (≥900 Hz), which may be necessary to encode sufficient amplitude modulations for speech recognition (**Loizou et al., 2000**). However, ITD discrimination has been shown to deteriorate when pulse rates exceeded a few hundred Hz in humans (**van Hoesel, 2007**; **Laback et al., 2007**) and animals (**Joris and Yin, 1998**; **Chung et al., 2016**). Our own behavioral experiments described here were conducted with low pulse rates (50 Hz), and future work will need to determine whether ITD discrimination performance declines at higher pulse rates which would make pulse rate an important factor for the development of good ITD sensitivity under this stimulation conditions. Thus, designers of novel bilateral CI speech processors may face conflicting demands: They must invent devices that fire each of 20 or more electrode channels in turn, at rates that are both fast, so as to encode the speech envelope in fine detail, but also slow, so as not to overtax the brainstem's ITD extraction mechanisms, and they must make the timing of at least some of these pulses encode stimulus fine structure and ITDs. While difficult, this may not be impossible, and promising, research is underway, which either provides fine structure information on up to four apical electrodes while running CIS strategy on the remaining electrodes (MED-EL CIs; **Riss et al., 2014**), uses a mixture of different pulse rates for different electrode channels (**Thakkar et al., 2018**), presents redundant temporal fine

structure information to multiple electrode channels (*Churchill et al., 2014*), or aims to 'reset' the brain's ITD extraction mechanisms by introducing occasional 'double pulses' into the stimulus (*Srinivasan et al., 2018*). A detailed discussion is beyond the scope of this article. Our results underscore the need to pursue this work with urgency as we have provided evidence that the absence of auditory input during a critical period does not necessarily mean that early deafened CI users show poor or no ITD sensitivity.

On a final note, we would be remiss if we did not acknowledge that, while the 'maladaptive plasticity hypothesis' that we have elaborated over the last few paragraphs is 'compatible' with the experimental data we have presented, it would be wrong to assert that our data so far prove that this hypothesis is correct. At present, we have merely managed to shed serious doubts on the popular critical period hypothesis, but at present, the maladaptive plasticity hypothesis is still, apart from others including different etiologies of deafness, just one possible alternative explanation for the observed poor ITD sensitivity of human bilateral CI users. It still needs to be put to the test by measuring the effect of deliberately degrading the quality of ITD cues to varying extent and over various periods. However, the animal model introduced in this study now makes this important task experimentally feasible.

# Materials and methods

All procedures involving experimental animals reported here were approved by the Department of Health of Hong Kong (#16–52 DH/HA and P/8/2/5) and/or the Regierungspräsidium Freiburg (#35–9185.81/G-17/124), as well as by the appropriate local ethical review committee. A total of 14 rats were obtained for this study from the breeding facilities of the Chinese University of Hong Kong or from Janvier Labs (Le Genest-Saint-Isle, France), and these were allocated randomly to the deafened and hearing experienced cohorts described in *Figure 1*.

## Deafening

Rats were deafened by daily intraperitoneal (i.p.) injections of 400 mg/kg kanamycin from postnatal days 9 to 20 inclusively (*Osako et al., 1979*; *Rosskothen-Kuhl and Illing, 2012*). This is known to cause widespread death of inner and outer hair cells (*Osako et al., 1979*; *Matsuda et al., 1999*; *Argence et al., 2008*) while keeping the number of spiral ganglion cells comparable to that in untreated control rats (*Osako et al., 1979*; *Argence et al., 2008*). *Osako et al., 1979* have shown that rat pups treated with this method achieve hearing thresholds around 70 dB for only a short period (~p14–16) and are severely to profoundly hearing impaired thereafter, resulting in widespread disturbances in the histological development of their central auditory pathways, including a nearly complete loss of tonotopic organization (*Rosskothen-Kuhl and Illing, 2012*; *Rauch et al., 2016*; *Rosskothen-Kuhl et al., 2018*). We verified that this procedure provoked profound hearing loss (>90 dB) by the loss of Preyer's reflex (*Jero et al., 2001*), the absence of ABRs to broadband click stimuli (*Figure 5b*) as well as pure tones (at 500, 1000, 2000, and 8000 Hz), and by performing histological assessment on cochlea sections of 11 weeks old, ND rats (data not shown). ABRs were measured as described in *Rosskothen-Kuhl et al., 2018* under ketamine (80 mg/kg) and xylazine (12 mg/kg) anesthesia each ear was stimulated separately through hollow ear bars with 0.5 ms broadband clicks with peak amplitudes up to 130 dB sound pressure level (SPL) delivered at a rate of 43 Hz. ABRs were recorded by averaging scalp potentials measured with subcutaneous needle electrodes between mastoids and the vertex of the rat's head over 400 click presentations. While normal rats typically exhibited click ABR thresholds near 30 dB SPL (*Figure 5a*), deafened rats had very high click thresholds of ≥130 dB SPL (*Figure 5b*).

## CI implantation, stimulation, and testing

All animals were implanted in early adulthood (between 10 and 14 weeks postnatally) for both behavioral training and electrophysiology recordings (*Figure 1*). All surgical procedures, including CI implantation and craniotomy, were performed under anesthesia induced with i.p. injection of ketamine (80 mg/kg) and xylazine (12 mg/kg). For maintenance of anesthesia during electrophysiological recordings, a pump delivered an i.p. infusion of 0.9% saline solution of ketamine (17.8 mg/kg/h) and xylazine (2.7 mg/kg/h) at a rate of 3.1 ml/h. During the surgical and experimental procedures, the body temperature was maintained at 38°C using a feedback-controlled heating pad (RWD Life

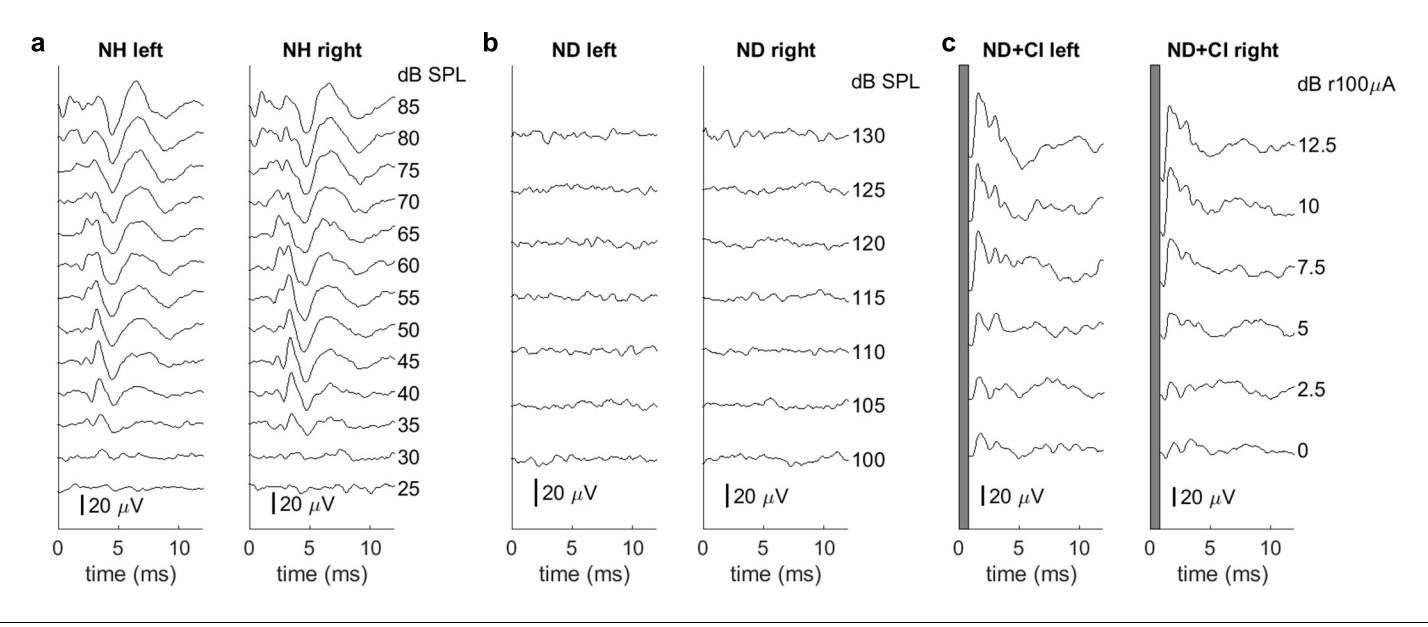

**Figure 5.** Examples of brainstem recordings to verify normal hearing or loss of hearing function as well as the symmetrical placement of cochlear implants (CIs). Each recording is from a single animal. Panels (**b**) and (**c**) come from the same animal pre- and post-CI implantation. (**a**) Auditory brainstem responses of an acoustically stimulated normal hearing (NH) rat. ABRs are symmetric for both ears and show clear differentiation. (**b**) ABRs of a neonatally deafened (ND) rat. No hearing thresholds were detectable up to 130 dB SPL. (**c**) Electrically evoked ABRs under CI stimulation of a deafened rat. Each sub-panel includes measurements for the left and the right ears, respectively, under acoustic (**a, b**) or electric stimulation (**c**). In (**c**), the first millisecond (electrical stimulus artifact) is blanked out.

Sciences, Shenzhen, China). Further detailed descriptions of our cochlear implantation methods can be found in previous studies (*Rosskothen et al., 2008*; *Rosskothen-Kuhl and Illing, 2010*; *Rosskothen-Kuhl and Illing, 2012*; *Rosskothen-Kuhl and Illing, 2014*; *Rosskothen-Kuhl and Illing, 2015*).

In short, two to four rings of an eight-channel electrode carrier (Cochlear Ltd animal array ST08.45, Cochlear Ltd, Peira, Beerse, Belgium) were fully inserted through a cochleostomy in medio-dorsal direction into the middle turn of both cochleae. Electrically evoked ABRs (EABRs) were measured for each ear individually to verify that both CIs were successfully implanted and operated at acceptably low electrical stimulation thresholds, usually around 100 µA with a duty cycle of 61.44 µs positive, 40.96 µs at zero, and 61.44 µs negative (*Figure 5c*). EABR recording used isolated biphasic pulses (see below) with a 23 ms inter-pulse interval. EABR mean amplitudes were determined by averaging scalp potentials over 400 pulses for each stimulus amplitude. For electrophysiology experiments, EABRs were also measured immediately before and after IC recordings, and for the chronically implanted rats, EABRs were measured once a week under anesthesia to verify that the CIs functioned properly and stimulation thresholds were stable.

## Electric and acoustic stimuli

The electrical stimuli used to examine the animals' EABRs, the physiological, and the behavioral ITD sensitivity were generated using a Tucker-Davis Technology (TDT, Alachua, FL) IZ2MH programmable constant current stimulator at a sample rate of 48,828.125 Hz. The most apical ring of the CI electrode served as stimulating electrode, the next ring as ground electrode. All electrical intracochlear stimulation used biphasic current pulses similar to those used in clinical devices (duty cycle: 61.44 µs positive, 40.96 µs at zero, 61.44 µs negative), with peak amplitudes of up to 300 µA, depending on physiological thresholds or informally assessed behavioral comfort levels (rats will scratch their ears frequently, startle or show other signs of discomfort if stimuli are too intense). For behavioral training, we stimulated all NDCI-B rats 6 dB above these thresholds.

Calibration measurements for electric ITD stimuli were performed by connecting the stimulator cable to 10 kΩ resistors instead of the in vivo electrodes and recording voltages using a Tektronix

MSO 4034B oscilloscope with 350 MHz and 2.5 GS/s. The stimulator was programmed to produce biphasic rectangular stimulus pulses with a 20 µA amplitude (y-axis) and a 20.5 µs interval between the positive and the negative phases. Measured calibration pulses such as those shown in *Figure 2—figure supplement 1c* were used to verify that electric ILDs were negligible and did not vary systematically with ITD. ILDs were computed as the difference in root mean square power of the signals in *Figure 2—figure supplement 1d*. These residual ILDs produced by device tolerances in our system are not only an order of magnitude smaller than the ILD thresholds for human CI subjects reported in the literature (~0.1 dB; *van Hoesel and Tyler, 2003*), they also do not covary with ITD. We can therefore be certain that sensitivity to ILDs cannot account for our behavior data. Acoustic stimuli used to measure behavioral ITD sensitivity in NH-B rats consisted of a single sample pulse (generated as a digital delta function 'click') at a sample rate of 48,000 Hz. Acoustic stimuli were presented via a Raspberry Pi three computer connected to a USB sound card (StarTech.com, Ontario Canada, part # ICUSBAUDIOMH), amplifier (Adafruit stereo 3.7W class D audio amplifier, New York City, NY, part # 987), and miniature high-fidelity headphone drivers (GQ-30783–000, Knowles, Itasca, IL), which were mounted on hollow tubes. The single sample pulse stimuli resonated in the tube phones to produce acoustic clicks, which decayed exponentially over a couple of millisecond (see *Figure 2—figure supplement 2d*). Stimuli were delivered at sound intensities of ≈80 dB SPL. A 3D printed 'rat acoustical manikin' with miniature microphones in each ear canal was used for validating that the acoustic setup delivered the desired ITDs and no usable intensity cues (see *Figure 2—figure supplement 2* and *Li et al., 2019*). Note that the residual ILDs are much smaller than the reported behavioral thresholds for ferrets (~1.3 dB *Keating et al., 2014*) or rats (~3 dB *Wesolek et al., 2010*). We can therefore be certain that sensitivity to ILDs cannot account for our behavior data.

To produce electric or acoustic stimuli of varying ITDs spanning the rat's physiological range of ±130 µs (*Koka et al., 2008*), stimulus pulses of identical shape and amplitude were presented to each ear, with the pulses in one ear delayed by an integer number of samples. Given the sample rates of the devices used, ITDs could thus be varied in steps of 20.48 µs for the electrical stimuli and 20.83 µs for the acoustic stimuli.

## Animal psychoacoustic testing

We trained our rats on 2AFC sound lateralization tasks using methods similar to those described in *Walker et al., 2009*; *Bizley et al., 2013*; *Keating et al., 2013a*; *Li et al., 2019*. The behavioral animals were put on a schedule with 6 days of testing, during which the rats obtained their drinking water as a positive reinforcer, followed by 1 day off, with ad lib water. The evening before the next behavioral testing period, drinking water bottles were removed. During testing periods, the rats were given two sessions per day. Each session lasted 25–30 min, which typically took 150–200 trials during which ≈10 ml of water were consumed.

One of the walls of each behavior cage was fitted with three brass water spouts, mounted ≈6–7 cm from the floor, and separated by ≈7.5 cm (*Figure 2—figure supplement 1a-b*; *Figure 2—figure supplement 2a-c*). We used one center 'start spout' for initiating trials and one left and one right 'response spout' for indicating whether the stimulus presented during the trial was perceived as lateralized to that side. Contact with the spouts was detected by capacitive touch detectors (Adafruit Industries, New York City, NY, part # 1362). Initiating a trial at the center spout triggered the release of a single drop of water through a solenoid valve. Correct lateralization triggered three drops of water as positive reinforcement. Incorrect responses triggered no water delivery but caused a 5–15 s timeout during which no new trial could be initiated. Timeouts were also marked by a negative feedback sound for the NH-B rats. Given that CI stimulation can be experienced as effortful by human patients (*Perreau et al., 2017*), and to avoid potential discomfort from prolonged negative feedback stimuli, the NDCI-B rats received a flashing LED as an alternative negative feedback stimulus. The LED was housed in a sheet of aluminum foil both to direct the light forwards and to ground the light to the setup. After each correct trial a new ITD was chosen randomly from a set spanning ±160 µs in 25 µs steps, but after each incorrect trial, the last stimulus was repeated in a 'correction trial'. Correction trials prevent animals from developing idiosyncratic biases favoring one side (*Walker et al., 2009*; *Keating et al., 2014*), but since they could be answered correctly without attention to the stimuli by a simple 'if you just made a mistake, change side' strategy, they are excluded from the final psychometric performance analysis.

The NH-B rats received their acoustic stimuli through stainless steel hollow ear tubes placed such that, when the animal was engaging the start spout, the tips of the tubes were located right next to each ear of the animal to allow near-field stimulation (*Figure 2—figure supplement 2a*). The pulses resonated in the tubes, producing pulse-resonant sounds, resembling single-formant artificial vowels with a fundamental frequency corresponding to the 50 Hz click rate. Note that this mode of sound delivery is thus very much like that produced by 'open' headphones, such as those commonly used in previous studies on binaural hearing in humans and animals, for example (*Wightman and Kistler, 1992*; *Keating et al., 2013a*). We used a 3D printed 'rat acoustical manikin' with miniature microphones in the ear canals (*Figure 2—figure supplement 2c*). It produced a channel separation between ears of $\geq$20 dB at the lowest, fundamental frequency, and around 40 dB overall. Further details on the acoustic setup and procedure are described in *Li et al., 2019*. The NDCI-B rats received their auditory stimulation via bilateral CIs described above, connected to the TDT IZ2MH stimulator via a custom-made, head mounted connector and commutator, as described in *Rosskothen-Kuhl and Illing, 2014*.

## Multi-unit recording from IC

Immediately following bilateral CI implantation, anesthetized NDCI-E rats were head fixed in a stereotactic frame (RWD Life Sciences), craniotomies were performed bilaterally just anterior to lambda. A single-shaft, 32-channel silicon electrode array (ATLAS Neuroengineering, E32-50-S1-L6) was inserted stereotactically into the left or right IC through the overlying occipital cortex using a micromanipulator (RWD Life Sciences). Extracellular signals were sampled at a rate of 24.414 kHz with a TDT RZ2 with a NeuroDigitizer head-stage and BrainWare software. Our recordings typically exhibited short response latencies ($\approx$ 3–5 ms), which suggests that they may come predominantly from the central region of IC. Responses from non-lemniscal sub-nuclei of IC have been reported to have longer response latencies ($\approx$ 20 ms; *Syka et al., 2000*).

At each electrode site, we first measured neural rate/level functions, varying stimulation currents in each ear to verify that the recording sites contained neurons responsive to cochlear stimulation, and to estimate threshold stimulus amplitudes. Thresholds rarely varied substantially from one recording site to another in any one animal. We then measured ITD tuning curves by presenting single pulse binaural stimuli with equal amplitude in each ear, $\approx$10 dB above the contralateral ear threshold, in pseudo-random order. ITDs varied from 163.84 μs contralateral ear leading to 163.84 μs ipsilateral ear leading in 20.48 μs steps. Each ITD value was presented 30 times at each recording site. The inter-stimulus interval was 500 ms. At the end of the recording session, the animals were overdosed with pentobarbitone.

## Data analysis

To quantify the extracellular multi-unit responses, we calculated the average activity for each stimulus over a response period (3–80 ms post-stimulus onset) as well as baseline activity (300–500 ms after stimulus onset) at each electrode position. The first 2.5 ms post-stimulus onset was dominated by electrical stimulus artifacts and were discarded. For display purposes of the raster plots in *Figure 3*, we extracted multi-unit spikes by simple threshold crossings of the band-passed (300 Hz–6 kHz) electrode signal with a threshold set at 4 standard deviation of the signal amplitude. To quantify responses for tuning curves, instead of counting spikes by threshold crossings, we instead computed an analog measure of multi-unit activity (AMUA) amplitudes as described in *Schnupp et al., 2015*. The mean AMUA amplitude during the response and baseline periods was computed by band-passing (300 Hz–6 kHz), rectifying (taking the absolute value) and low-passing (6 kHz) the electrode signal. This AMUA value thus measures the mean signal amplitude in the frequency range in which spikes have energy. As illustrated in Figure 1 of *Schnupp et al., 2015*, this gives a less noisy measure of multi-unit neural activity than counting spikes by conventional threshold crossing measures because the latter are subject to errors due to spike collisions, noise events, or small spikes sometimes reach threshold and sometimes not. The tuning curves shown in the panels of *Figure 3* are the normalized responses from this AMUA measure averaged across 30 trials for each ITD seen by each of the dots per vertical panel in the raster plots where each panel is an ITD and each dot is a spike. Changes in the AMUA amplitudes tracked changes in spike density.

## STVR calculation

STVR values are a measure of the strength of tuning of neural responses to ITD, which we adopted from *Hancock et al., 2010* to facilitate quantitative comparisons. The STVR is defined in *Hancock et al., 2010* as the proportion of trial-to-trial variance in response amplitude explained by changes in ITD. The STVR is calculated by computing a one-way ANOVA of responses grouped by ITD value and dividing the total sum of squares by the group sum of squares. This yields values between 0 (no effect of ITD) and 1 (response amplitudes completely determined by ITD). p-Values were also computed from the one-way ANOVA and p<0.01 served as a criterion to determine whether the ITD tuning of a given multi-unit was deemed statistically significant. The number of responses for each ITD value was 30, yielding with a degree of freedom for the ANOVA of 29.

## Psychometric curve fitting

In order to derive summary statistics that could serve as measures of ITD sensitivity from the thousands of trials performed by each animal, we fitted psychometric models to the observed data. It is common practice in human psychophysics to fit performance data with cumulative Gaussian functions (*Wickens, 2002*; *Schnupp et al., 2005*). This practice is well motivated in signal detection theory, which assumes that the perceptual decisions made by the experimental subject are informed by sensory signals, which are subject to multiple, additive, and hence approximately normally distributed sources of noise. When the sensory signals are very large relative to the inherent noise, then the task is easy and the subject will make the appropriate choice with near certainty. For binaural cues closer to threshold, the probability of choosing the 'right' spout ($p_R$) can be modeled by the function

$$p_R = \Phi(ITD \cdot \alpha) \tag{1}$$

where $\Phi$ is the cumulative normal distribution, *ITD* denotes the interaural time difference (arrival time at left ear minus arrival time at right ear, in ms), and $\alpha$ is a sensitivity scale parameter that captures how big a change in the proportion of 'right' choices a given change in ITD can provoke, with units of 1/ms.

Functions of the type in *Equation 1* tend to fit psychometric data for 2AFC tests with human participants well, where subjects can be easily briefed and lack of clarity about the task, lapses of attention, or strong biases in the perceptual choices are small enough to be explored. However, animals have to work out the task for themselves through trial and error and may spend some proportion of trials on 'exploratory guesses' rather than direct perceptual decisions. If we denote the proportion of trials during which the animal makes such guesses (the 'lapse rate') by $\gamma$, then the proportion of trials during which the animal's responses are governed by processes, which are well modeled by *Equation 1*, is reduced to $(1-\gamma)$. Furthermore, animals may exhibit two types of bias: an 'ear bias' and a 'spout bias'. An 'ear-bias' exists if the animal hears the midline (50% right point) at ITD values that differ from zero by some small value $\beta$. A 'spout bias' exists if the animal has an idiosyncratic preference for one of the two response spouts or the other, which may increase its probability of choosing the right spout by $\delta$ (where $\delta$ can be negative if the animal prefers the left spout). Assuming the effect of lapses, spout, and ear bias to be additive, we therefore extended *Equation 1* to the following psychometric model:

$$p_R = \Phi(ITD \cdot \alpha + \beta) \cdot (1 - \gamma) + \frac{\gamma}{2} + \delta \tag{2}$$

We fitted the model in *Equation 2* to the observed proportions of 'right' responses as a function of stimulus ITD using the scipy.optimize.minimize() function of Python 3.4, using gradient descent methods to find maximum likelihood estimates for the parameters $\alpha$, $\beta$, $\gamma$, and $\delta$ given the data. This cumulative Gaussian model fitted the data very well, as is readily apparent in *Figure 2a–j*. We then used the slope of the psychometric function around zero ITD as our maximum likelihood estimate of the animal's ITD sensitivity, as plotted in *Figure 2k*. That slope is easily calculated using the *Equation 3*

$$slope = \varphi(0) \cdot \alpha \cdot (1 - \gamma) \tag{3}$$

which is obtained by differentiating *Equation 1* and setting ITD = 0. $\varphi(0)$ is the Gaussian normal probability density at zero ($\approx 0.3989$).

Seventy-five percent of correct thresholds were computed as the mean absolute ITD at which the psychometric dips below 25% or rises above 75% 'right' responses, respectively.

## Acknowledgements

We thank A Hyun Jung for assisting behavioral training of CI rats, P Ruther and the *Cluster of Excellence BrainLinks-BrainTools* (German Research Foundation, grant number EXC1086) for the support with recording electrodes. Work leading to this publication was supported by grants from the Hong Kong General Research Fund (11100219) and Medical Research Fund (06172296), the Shenzhen Science and Innovation Fund (JCYJ20180307124024360), the German Academic Exchange Service (DAAD) with funds from the German Federal Ministry of Education and Research (BMBF) and the People Programme (Marie Curie Actions) of the European Union's Seventh Framework Programme (FP7/2007-2013) under REA grant agreement n° 605728 (PRIME – Postdoctoral Researchers International Mobility Experience), and friends' association 'Taube Kinder lernen hören e V'. The article processing charge was funded by the Baden-Wuerttemberg Ministry of Science, Research and Art and the University of Freiburg in the funding programme Open Access Publishing.

## Additional information

### Funding

| Funder | Grant reference number | Author |
|---|---|---|
| Hong Kong Government General Research Fund (GRF) | 11100219 | Jan WH Schnupp |
| Friends Association "Taube Kinder lernen hören e.V." | | Nicole Rosskothen-Kuhl |
| Hong Kong Health and Medical Research Fund (HMRF) | 06172296 | Jan WH Schnupp |
| Shenzhen Science and Innovation Fund | JCYJ20180307124024360 | Jan WH Schnupp |
| German Academic Exchange Service | 605728 (PRIME – Postdoctoral Researchers International Mobility Experience) | Nicole Rosskothen-Kuhl |
| Deutsche Forschungsgemeinschaft | Grant number EXC1086, Cluster of ExcellenceBrainLinks-BrainTools | Nicole Rosskothen-Kuhl |
| Ministerium für Wissenschaft, Forschung und Kunst Baden-Württemberg | | Nicole Rosskothen-Kuhl |
| Universität Freiburg | Funding programme for Open Access Publishing | Nicole Rosskothen-Kuhl |

The funders had no role in study design, data collection and interpretation, or the decision to submit the work for publication.

### Author contributions

Nicole Rosskothen-Kuhl, Jan WH Schnupp, Conceptualization, Resources, Data curation, Software, Formal analysis, Supervision, Funding acquisition, Validation, Investigation, Visualization, Methodology, Writing - original draft, Project administration, Writing - review and editing; Alexa N Buck, Data curation, Software, Formal analysis, Validation, Investigation, Visualization, Methodology, Writing - original draft, Writing - review and editing; Kongyan Li, Methodology

## Author ORCIDs

Nicole Rosskothen-Kuhl https://orcid.org/0000-0003-4724-5550
Alexa N Buck https://orcid.org/0000-0003-0124-9716

## Ethics

Animal experimentation: All procedures involving experimental animals reported here were approved by the Department of Health of Hong Kong (#16-52 DH/HA&P/8/2/5) or Regierungspräsidium Freiburg (#35-9185.81/G-17/124), as well as by the appropriate local ethical review committee. All surgery was performed under ketamine and xylazine anesthesia, and every effort was made to minimize suffering.

## Decision letter and Author response

Decision letter https://doi.org/10.7554/eLife.59300.sa1
Author response https://doi.org/10.7554/eLife.59300.sa2

# Additional files

## Supplementary files

• Transparent reporting form

## Data availability

All data generated or analysed during this study are included in the manuscript and supporting files. Data have been deposited to Dryad, under the https://doi.org/10.5061/dryad.573n5tb6d .

The following dataset was generated:

| Author(s) | Year | Dataset title | Dataset URL | Database and Identifier |
|---|---|---|---|---|
| Rosskothen-Kuhl N, Buck AN, Li K, Schnupp JW | 2021 | Behavioral and ephys data of research paper | https://datadryad.org/stash/share/H1O6hxXk-WYe_aw0RyRkqaYa-gISFntiESDG9qNjxUrrc | Dryad Digital Repository, 10.5061/dryad.573n5tb6d |

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
