## [Decision Letter]

**Acceptance summary:**

This is the first study to demonstrate near-normal interaural time difference (ITD) thresholds using both behavioral tasks and acute electrophysiological recordings in an early-deafened animal model. This finding contrasts with the poor ITD sensitivity in early-deafened human patients with bilateral cochlear implants, suggesting that deafness and atrophy of neural pathways may not by themselves cause poor ITD sensitivity. This raises the possibility that in human patients, other factors such as exposure to bilateral stimulation with unsynchronized CI devices may cause maladaptive plasticity and interfere with the ability to use ITD cues.

**Decision letter after peer review:**

Thank you for submitting your article "Microsecond Interaural Time Difference Discrimination Restored by Cochlear Implants After Neonatal Deafness" for consideration by *eLife*. Your article has been reviewed by three peer reviewers, including Lina Reiss as the Reviewing Editor and Reviewer #1, and the evaluation has been overseen by Barbara Shinn-Cunningham as the Senior Editor.

The reviewers have discussed the reviews with one another and the Reviewing Editor has drafted this decision to help you prepare a revised submission.

As the editors have judged that your manuscript is of interest, but as described below that additional experiments may be required before it is published, we would like to draw your attention to changes in our revision policy that we have made in response to COVID-19 (https://elifesciences.org/articles/57162). First, because many researchers have temporarily lost access to the labs, we will give authors as much time as they need to submit revised manuscripts. We are also offering, if you choose, to post the manuscript to bioRxiv (if it is not already there) along with this decision letter and a formal designation that the manuscript is "in revision at *eLife*". Please let us know if you would like to pursue this option. (If your work is more suitable for medRxiv, you will need to post the preprint yourself, as the mechanisms for us to do so are still in development.)

Summary:

All three reviewers were generally positive and in favor of publication after attention to a large number of specific details. The reviewers also noted and agreed that the conclusions need to be toned down regarding the role of maladaptive stimulation, which received a great deal of attention in the manuscript, but is not sufficiently supported by the existing data. There are additional experiments that would need to be done to support that claim, which are beyond the scope of the current manuscript. While this should be toned down, the findings do raise important questions about the role of maladaptive plasticity, and more clarity is needed in the discussion of the roles of deafness versus experience in the development of ITD sensitivity.

In addition the reviewers felt that results are quite surprising given the previous findings in similar animal studies. More details of the current experiments and discussion of and comparison to previous published work in animals and humans are needed to understand the differences in the findings. For instance, more details on animal groups as noted by reviewer 2 and consideration of the role of implantation in the middle rather than basal turn of the cochlea, as well as the roles of human etiologies noted by reviewer 1.

In addition, more precise discussion of previous literature on ITD sensitivity is needed as noted by reviewer 2, as well as a thorough discussion of critical period and the relationship to ITD sensitivity as noted by reviewer 3. The authors should also clearly state the hypotheses of the study as noted by reviewer 3.

Reviewer #1:

This is an interesting study that examines whether neonatally deafened animals can learn to use ITD cues provided by bilateral cochlear implants.

This is the first study to investigate this with animal training as well as acute electrophysiological recordings as previously done in cats and rabbits. The finding of near-normal ITD sensitivity in these animals, which contrasts with findings in human patients with bilateral cochlear implants, raises important questions about the role of maladaptive plasticity due to previous hearing aid or unsynchronized CI use, as opposed to just atrophy due to deafness alone. At the very least, the findings suggest that ITD coding is possible to restore even if deafening occurred early in the critical period.

Another way that this study differs from the previous animal studies is implantation of the electrode array in the middle rather than basal turn of the cochlea, which may allow current to better reach lower frequency neurons that are better represented in ITD pathways. This may also explain the improved ITD coding observed in the ICC compared to the previous findings in cats.

While the proposed role of maladaptive plasticity is compelling, additional proof would be needed to confirm this, such as the converse experiment in which ITD sensitivity is measured after exposure to unsynchronized pulse trains. There are other explanations for poor ability to use ITD cues in humans, including different etiologies of deafness in those deafened as children (usually genetic, not chemical), which may affect neural coding and synchronization. Some people with congenital deafness have zero auditory experience, unlike the rats in this study which have some minimal auditory experience before deafening. Thus, the converse experiment is necessary to address these potential alternative explanations, and should be discussed.

Some additional methodology concerns:

1) ABRs were measured using clicks, not tones. For rigor, can you provide data from one animal using tones for ABR, to verify complete hearing loss at all frequencies, especially low frequencies which are less vulnerable to chemical deafening? Verification is more important for this study than for the previous studies which placed the implant in the basal part of the cochlea, near the region of deafening.

2) The level differences in Figure 2—figure supplement 1D show that there was a higher ILD for +100 than -100 microseconds. Why did this occur, and why wasn't this cue eliminated? This leaves open the possibility that rats could use this cue instead of ITD. Please also show the calibration results for the intermediate ITDs, not just -100 and +100.

3) Why were the NH rats stimulated with two tubes instead of a single loudspeaker? Wouldn't this lead to something like the precedence effect, and complicate interpretation of the NH data? The rationale should be discussed (I presume this is to control for ILD).

Reviewer #2:

This manuscript reports behavioral and physiological sensitivity of neonatally deafened rats to interaural timing differences (ITD) in cochlear implant stimulation. The reported thresholds, around 50 microsec, are remarkably short, certainly shorter than observations in pre-lingually deaf humans and considerably shorter than most physiological results in other animals. Indeed, I was initially inclined to question the result, but the experimental design and conduct of the experiments seem solid. The over-zealous tone of the presentation detracts from the impact of the study.

The authors tend to overstate the poor ITD sensitivity of cochlear implant (CI) users. In the Introduction, for instance, states that "ITD sensitivity of CI patients is poor or completely missing". This might be more-or- less true of pre-lingually deaf CI patients. It is not correct, however, for post-lingually deaf patients, and the authors don't make that distinction here. Figure 2 of Laback et al., 2011, for instance, shows that the median ITD threshold for post-lingual patients is between 100 and 200 microsec and that post-lingual patients with no ITD sensitivity are rare (about 9% in that figure). Thresholds of 100-200 microsec are comparable to those of normal hearing human listeners detecting interaural delays in sound envelopes. Again, in subsection “Early deaf CI rats discriminate ITD as accurately as their normally hearing litter mates”, I think that it is not correct that pre-lingual CI patients "usually" have ITD thresholds "too large to measure", although I would accept "often". It is important to note that ITD sensitivity in CI patients declines as pulse rates are increased above about 50 pps, and clinical processors run at >900 pps. The present study used single bilateral pulse pairs for physiology and relatively low (50 pps) pulse rates for behavior. For that reason, the present results in rats cannot be compared with human results obtained with clinical processors, only with those made with laboratory processors.

There are as many as 5 experimental groups of rats that are lumped together as neonatally deafened (ND) and normal hearing (NH). I count the following groups:

1) Neonatally deafened, trained and tested for behavior

2) Neonatally deafened, tested with inferior colliculus (IC) recording

3) Raised with normal hearing, trained and tested for behavior using sound stimulation

4) Raised with normal hearing, tested with IC recording and sound stimulation

5) Raised with normal hearing, deafened as adults (I think), implanted with ICs, and tested with IC recording – These animals are referred as "NH rats", the same as to groups 3 and 4.

We need to know if groups 1 and 2 are the same rats. If so, what was the history of electrical stimulation prior to IC recording? That is, did they get just 2 25-30-minute sessions a day of stimulation for about 14 days, or did they receive some other electrical stimulation? Were group 5 rats deafened, implanted, and recorded all the same day, or did time elapse between implantation and IC recording? Were those rats deafened and, if so, how? We need a different label to distinguish group 5 from groups 3 and 4. The caption of Figure 3 refers to "hearing experienced" rats and cats, which I think is a useful term. Still, the presentation of Figure 3 in the text refers only to "NH", and it is not obvious that the data in Figure 3 are, I guess, from rats that were raised with normal hearing, then deafened, implanted, and stimulated electrically as young adults.

Reviewer #3:

This paper has significance in the fields of Developmental Biology and Neuroscience and a warrants publication in *eLife*. The aim of the current study, using behavioral and physiological evidence, was to determine if interaural time difference (ITD) thresholds can be measured in neonatally deafened rats when implanted with cochlear implants in adulthood. In addition, the rats were provided with rigorous training to ITDs. A secondary aim of the study was to debunk the idea of an early "critical period" for learning ITDs. The study found that neonatally deafened rats were able to behaviorally respond and learn to use ITDs with exquisite precision even when no input is provided during an early "critical period" and that this was due to the presentation of consistent, salient ITDs after cochlear implantation.

As a basic science question, the work in this study provides an excellent animal model for future behavioral testing in binaural tasks. In addition, the methods and implementation are entirely sound and the work is well-motivated, and falls within the scope of this journal. Clinically, there is a need in the field to understand the impact of bilateral implantation on the developing auditory system and its ability to provide the necessary cues for sound localization. This study provides a good stepping stone for future developmental studies that would like to probe whether better processing strategies, or early intervention with cochlear implants, is the more pressing issue for ascertaining ITD sensitivity. However, there were major logical flaws or gaps in the manuscript. These could easily be fixed with changes to the language in the Introduction and Discussion. These major issues are outlined below:

1) The concept of the "critical period" is not well-defined. More of the literature on the relationship between critical periods and ITD sensitivity should be surveyed in order to understand what the authors are referring to in the context of the present study. First, it would be very helpful to the reader to be able to visualize, with a schematic or timeline, how the animals in the current study were, deafened, implanted, and subsequently trained with ITDs, compared with other studies. Second, what is this nature of this critical period? Is it a period for development of a skill, a vulnerable period, a period for recovery from deprivation? These examples are taken from Kral, 2013 as examples of previously published definitions. I think it might be in the authors' benefit to clearly define what this critical period represents so that the interpretation of the results can be easily understood.

2) Throughout the Introduction and Discussion there is an implicit suggestion about how deafness at an early age may not necessarily cause a loss of ITD sensitivity, and instead that CIs have limitations which could prevent learning of ITDs. However, a distinction needs to be made on what type of limitations are being debated: is it (a) the inability of CI to provide ITDs or is it the (b) exposure to inconsistent ITDs through bilateral CIs that is preventing learning of those cues? In the manuscript, the authors appear to advocate for the latter point. This distinction is further confused by the sentence in the Introduction, where the sensitivity of CI patients is mentioned, citing papers with a mix of measurements made with both clinical processors and synchronized research processors. Therefore, if the authors are suggesting that poor delivery of ITDs using unsynchronized processors is preventing learning of ITD cues in humans, then an equivalent condition with unsynchronized processors needs to be tested in the rats. This must be done in order to confidentially say that synchronized ITDs through the CI need to be conveyed in order to promote learning of binaural cues.

3) A final consideration, the authors should clearly state the hypotheses of the study. The authors have designed a unique study with two experiments, one behavioral and one physiological, whereby response lateralization and the proportion of ITD sensitive neurons were used as a metric to understand whether ND rats were able to (a) utilize ITDs and (b) possessed the underlying neural structures which reflect the behavioral data. It is imperative for the authors to state a set of hypotheses in relation to these two experiments and subsequently describe to the reader how they were carried out and measured. This means that, somewhere in the Introduction, there should be a justification for measuring response lateralization and proportion of ITD sensitive neurons/SNR for the behavioral and physiological experiments, respectively.

Introduction:

Needs a thorough literature review on the auditory critical period in relation to ITD sensitivity. The authors should also discuss why there hasn't been a rigorous study to understand critical periods and ITD sensitivity.

Needs to make distinction between early-deafened vs. late-deafened. This will help to understand the relation between human and rats.

Needs more information on training and plasticity in order to build up the question. While prior studies have shown that early deafness does impact ITD sensitivity, they do not suggest that sensitivity could not be relearned. Please describe evidence (if any) that supports the notion that lack of exposure to binaural cues in adulthood, rather than a total loss of binaural function early in life, limits sensitivity to binaural cues for early-deafened populations.

As stated earlier, a distinction should be made between synchronized delivery of binaural cues and unsynchronized bilateral implantation and which one of these is the contributing factor to improper learning or acquisition of ITDs in adulthood.

Subsection “Varying degrees and types of ITD tuning are pervasive in the neural responses in the IC of ND rats immediately after adult cochlear implantation”: The interpretation of various shapes of tuning curves aren't always known to the average reader. It would be helpful, either here or in the data analysis section, to describe what each of the curve types described about the neurons that are being recorded from.

Discussion: Please expand on the importance of ITD processing in rats and what "generally poor" means in the context of the current study.

Discussion: The logic here is a little hard to follow. The authors state that only 48% of IC neurons were found to be ITD sensitivity in ND subjects, conversely in the current study, 91% of IC neurons were ITD sensitive. The claim appears to be that because the proportion of ITD sensitive neurons was greater in the present study, that this is a hallmark of relearning of ITDs in the animal. However, nowhere in the Introduction or the hypotheses of the paper was this addressed. The hypotheses of paper need to be clearly stated such that the reader can understand what measures from the (a) behavioral and (b) physiological data constitute "good sensitivity to ITDs".

Discussion: This sentence, "It is important to remember that…congenital cats" is very long and a little difficult to follow. I think it would be helpful if the authors previously described what characterized good vs. poor tuning. It sounds like the authors are trying to make a connection between the behavioral data and the neural data, however, this is unclear.

Discussion: There is a reference to "critical periods" in each of these animal models, however, the statements made are rather vague. In each the ferret, mouse, and gerbil models, there is a reference to certain synapses and their relationship to binaural circuity prior to the onset of hearing (i.e. the "critical period"). However, the authors do not explain what happens to the circuitry, and what is required to occur (either before or after the so-called "critical period") in order to overcome any deficits in ITD processing. In addition, the authors have not explicitly stated what makes the rat model in present study unique for understanding the implications of the hypothesized "critical period."

Discussion: Authors should expand on why inhibition is important.

[Editors' note: further revisions were suggested prior to acceptance, as described below.]

Thank you for submitting your article "Microsecond Interaural Time Difference Discrimination Restored by Cochlear Implants After Neonatal Deafness" for consideration by *eLife*. Your article has been reviewed by three peer reviewers, including Lina Reiss as the Reviewing Editor and Reviewer #1, and the evaluation has been overseen by Barbara Shinn-Cunningham as the Senior Editor.

The reviewers have discussed the reviews with one another and the Reviewing Editor has drafted this decision to help you prepare a revised submission.

Summary:

The reviewers agree that the authors have made substantial improvements, especially with toning back the maladaptive plasticity hypothesis. However, there are two major remaining issues.

First, the writing of the Introduction still needs significant work, with more specific and crisper language, especially when referring to literature or ideas, such as when discussing the contribution of the critical period and related hypotheses as well as the role of training, as noted by reviewer 3. The authors also need to clarify interpretation of the rats' ability to use ITDs with exquisite precision as due to (1) provision of salient ITDs (independent of missed critical period) or (2) rigorous training and synchronized stimulation allowed rats to learn to use ITDs even though the critical period was missed.

There is also a new concern raised by reviewer 2 about the HE-CI group which the revision indicates were not chemically deafened before implantation. Specifically, there is a potential for electrophonic responses (e.g. recent work by Kral's group (Sato et al., 2016). The acoustic frequency corresponding to the parameters used in this paper would be approximately the period of the pulse, in this case 1/0.000164 or ~6000 Hz. It is not clear what tonotopic areas were recorded in the study and how electrophonic responses would affect the result. Even though the HE-CI group was given a conductive loss, this would not attenuate the signal as completely as chemical deafening, and inner/outer hair cells are likely to remain intact. and permit indirect electrophonic stimulation of inner hair cells via the outer hair cells, as well as direct electrical stimulation of inner hair cells. After discussion, all of the reviewers agreed that this is a substantial experimental confound, and also that this group does not add value to the study. Therefore, we recommend to remove this group.

---

## [Author Response]

Summary:All three reviewers were generally positive and in favor of publication after attention to a large number of specific details. The reviewers also noted and agreed that the conclusions need to be toned down regarding the role of maladaptive stimulation, which received a great deal of attention in the manuscript, but is not sufficiently supported by the existing data. There are additional experiments that would need to be done to support that claim, which are beyond the scope of the current manuscript. While this should be toned down, the findings do raise important questions about the role of maladaptive plasticity, and more clarity is needed in the discussion of the roles of deafness versus experience in the development of ITD sensitivity.In addition. the reviewers felt that results are quite surprising given the previous findings in similar animal studies. More details of the current experiments and discussion of and comparison to previous published work in animals and humans are needed to understand the differences in the findings. For instance, more details on animal groups as noted by reviewer 2 and consideration of the role of implantation in the middle rather than basal turn of the cochlea, as well as the roles of human etiologies noted by reviewer 1.In addition, more precise discussion of previous literature on ITD sensitivity is needed as noted by reviewer 2, as well as a thorough discussion of critical period and the relationship to ITD sensitivity as noted by reviewer 3. The authors should also clearly state the hypotheses of the study as noted by reviewer 3.

These are highly constructive criticism and we have substantially rewritten the Discussion to accommodate them. In particular that, while our results shed substantial doubts on the, at present prevailing, critical period hypothesis, they do not by themselves prove that the alternative maladaptive plasticity hypothesis is correct. We now end the Discussion on this important point. We have also clarified the various experimental cohorts, expanded the discussion of the human clinical literature that provides the background, and edited our Introduction as suggested by reviewer 3 to spell out our principal objective (that is, to attempt falsification of the critical period hypothesis) in the Introduction. We believe that we have been able to accommodate all the comments and criticisms of this excellent set of careful but constructive and fair minded reviews.

Reviewer #1:This is an interesting study that examines whether neonatally deafened animals can learn to use ITD cues provided by bilateral cochlear implants.This is the first study to investigate this with animal training as well as acute electrophysiological recordings as previously done in cats and rabbits. The finding of near-normal ITD sensitivity in these animals, which contrasts with findings in human patients with bilateral cochlear implants, raises important questions about the role of maladaptive plasticity due to previous hearing aid or unsynchronized CI use, as opposed to just atrophy due to deafness alone. At the very least, the findings suggest that ITD coding is possible to restore even if deafening occurred early in the critical period.Another way that this study differs from the previous animal studies is implantation of the electrode array in the middle rather than basal turn of the cochlea, which may allow current to better reach lower frequency neurons that are better represented in ITD pathways. This may also explain the improved ITD coding observed in the ICC compared to the previous findings in cats.While the proposed role of maladaptive plasticity is compelling, additional proof would be needed to confirm this, such as the converse experiment in which ITD sensitivity is measured after exposure to unsynchronized pulse trains. There are other explanations for poor ability to use ITD cues in humans, including different etiologies of deafness in those deafened as children (usually genetic, not chemical), which may affect neural coding and synchronization. Some people with congenital deafness have zero auditory experience, unlike the rats in this study which have some minimal auditory experience before deafening. Thus, the converse experiment is necessary to address these potential alternative explanations, and should be discussed.

We thank Lina Reiss for her constructive criticism of our manuscript. We agree that, at present, the “maladaptive plasticity hypothesis” we put forward is merely one of several possible explanations for the different outcomes in our rats and in human patients. This is now spelled out very clearly in the final paragraph of the Discussion. We also agree that "converse experiments" in which CI rats experience unsynchronized ITDs from the beginning of CI stimulation would be an essential follow-up, and we are already piloting such experiments.

Some additional methodology concerns:1) ABRs were measured using clicks, not tones. For rigor, can you provide data from one animal using tones for ABR, to verify complete hearing loss at all frequencies, especially low frequencies which are less vulnerable to chemical deafening? Verification is more important for this study than for the previous studies which placed the implant in the basal part of the cochlea, near the region of deafening.

We have recorded pure tone ABRs as best we can with our current setup as requested by the reviewer, for one ND rat and one normally hearing control rat. The results are shown in Author response image 1. Please be aware that our current setup uses headphone drivers which are not suitable for low frequency stimulation at high amplitudes. Signals will distort badly at high levels and low frequencies, which is why the recordings shown here only include sound levels up to 80 dB SPL. We are aware that ideally one would test substantially louder sound levels if one wanted to be absolutely sure that the animals involved are not just moderately but severely hearing impaired across the entire frequency region, even at frequencies below 1 kHz. To achieve that we would need to retool and recalibrate our setup with low frequency transducers, and if the reviewer insists that we do so, we will, but we hope to persuade the reviewer that this should not be necessary for two reasons: Firstly, since all our behavior results are obtained with electrodes implanted in the 8 kHz region, so even if there was any residual low frequency hearing, we would have to assume that this somehow induces essentially perfect ITD sensitivity in severely deaf frequency bands a full four octaves away, which would be extremely surprising. Secondly, and even more importantly, we have recently conducted histological studies on rats deafened neonatally with our kanamycin protocol and we were able to show that the treatment leads to a complete degeneration of the organ of corti along the entire length of the cochlea. This histological evidence is shown in Author response image 1, and it precludes any possibility of residual hearing in our ND animals more effectively than ABR recordings ever could. We hope that, together, these lines of evidence will reassure the reviewer that residual low frequency hearing can be safely disregarded as an explanation for the excellent CI ITD sensitivity documented in our manuscript.

**Author response image 1. sa2fig1:** Pure tone ABRs for one normally hearing (NH, top row) and one neonatally deafened (ND, bottom row) animal.

The histological assessment of the effect of the neonatal kanamycin treatment was performed as follows: we perfused and cut the cochleae of 11 weeks old (time of bilateral CI implantation) normal hearing (n=5) and neonatally deafened (n=3) rats and visualized the four Organs of Corti (basal turn, lower middle turn, upper middle turn, apical turn) by using hematoxylin and eosin staining over all 2.5 turns of the rat cochlea. In comparison to the Organ of Corti of normal hearing rats, we observed a complete loss of outer and inner hair cells over all turns of neonatally deafened rats (see Author response image 2, second row). A residual hearing ability of these animals in the low-frequency range can thus be excluded.

**Author response image 2. sa2fig2:** Organs of Corti of 11 weeks old normal hearing (first row) and neonatally deafened (second row) rats. All four Organs of Corti from base to apex are shown from left to right. While the Organs of Corti of normal hearing rats show three rows of outer hair cells (green arrow) and one row of inner hair cells (orange arrow), they are completely missing in all cochlea turns of the deafened rats. Scale bar: 20 µm.

2) The level differences in Figure 2—figure supplement 1D show that there was a higher ILD for +100 than -100 microseconds. Why did this occur, and why wasn't this cue eliminated? This leaves open the possibility that rats could use this cue instead of ITD. Please also show the calibration results for the intermediate ITDs, not just -100 and +100.

As requested by the reviewer, we have collected calibration data for additional ITD values and replaced the original Figure 2—figure supplement 1.

There are very small, artefactual ILDs observed in Figure 2—figure supplement 1D which are attributable to a tiny amount of capacitive/inductive channel crosstalk in the wires leading from the programmable stimulator to the implants. A current pushed through one wire will induce a tiny current in the wire running parallel to it by magnetic induction. If you look carefully at the traces in Figure 2C, you can see tiny little red bumps coinciding with big blue rising or falling phases and vice versa, corresponding to these induced currents. You see these bumps at the rising and falling phases because magnetic induction of currents is proportional to rate of change in field strength. The currents measured by the oscilloscope and used here for stimulus calibration are thus a superposition of the direct stimulus current injected into a given channel, plus the very much smaller induced current from the cross-talk from the neighboring channel. The direct current pulses and the cross talk current pulses can be either in phase or out of phase with each other depending on the ITD, which will lead to either constructive or destructive interference. This accounts for the small and variable ITD-induced ILD observed.

Why wasn’t this cue eliminated? Because doing so would be both hard and not worth the effort. Using higher quality shielded cables to reduce inductive coupling is only possible for part of the assembly if we want to keep the cable to the head connector manageably light and flexible. Reducing the rising slope of the pulses in order to reduce the amplitude of the magnetic induction would have led to pulse shapes that would have been atypical of those used in clinical practice. One could try to compute predictive inverse filters to model and subtract at source the magnetic induction currents in the wires, which would have to be done to very high accuracy to compensate for such small effects, but the value of the considerable effort involved would be nil given that the induced ILD “cue” is not only much smaller than the ILD threshold of rats. At 100 μs ITD, where our rats routinely achieve 80% correct or better (compare Figure 2) the ILD is as low as (-)0.018 dB and does not change sign with the ITD. The largest absolute ITD-induced ILD is 0.18 dB, or equivalently 2.17%. This is roughly an order of magnitude smaller than the rats ILD threshold. To illustrate this, the new Figure 2—figure supplement 1 contains a new panel E which shows behavioral ILD psychometric curves obtained from two additional ND-CI rats currently being tested for follow-on studies. Note that the ITD-induced ILDs shown in d also lack the systematic, monotonic relationship with ITD that would be necessary to explain away the remarkable ITD sensitivity observed in our ND-CI animals. We hope the reviewer can agree that the possibility that the animals could have used these tiny sub-threshold ILDs which lack a systematic relationship with ITD can be firmly excluded.

3) Why were the NH rats stimulated with two tubes instead of a single loudspeaker? Wouldn't this lead to something like the precedence effect, and complicate interpretation of the NH data? The rationale should be discussed (I presume this is to control for ILD).

In contrast to the study by e.g. Wesoleck et al., (2010), who presented open field sounds from loudspeakers, we were interested in presenting the sounds dichotically. A single loudspeaker would give very poor control over the actual ITDs, given the hard to quantify and control reverberation of open field sound in the test chamber. Dichotic stimulation not only gives better stimulus control, but also more closely reflects the situation in CI stimulation, where each ear is stimulated essentially independently. The ear tubes we devised for our near-field acoustic set up (see Figure 2—figure supplement 2A) achieves this by operating like a pair of “open” stereo headphones, such as those commonly used in previous studies of binaural hearing in humans and animals, (e.g. Keating et al., 2013). We validated this close field, open headphone stimulation method with a 3D printed “rat Kemar” dummy head, as shown in Figure 2—figure supplement 2D and described in further detail in Li et al., (2019). This enabled us to verify that the acoustic ITDs at the rats ears correspond faithfully to the ITDs imposed on the electrical signals sent to the headphone drivers. The precedence effect does not enter into this any more than it would with any study delivering binaural stimulation over open headphones. We have edited the main text and the legend of Figure 2—figure supplement 2 to indicate that these sound tubes operate like a set of open stereo headphones. This will hopefully clarify the rationale.

Reviewer #2:This manuscript reports behavioral and physiological sensitivity of neonatally deafened rats to interaural timing differences (ITD) in cochlear implant stimulation. The reported thresholds, around 50 microsec, are remarkably short, certainly shorter than observations in pre-lingually deaf humans and considerably shorter than most physiological results in other animals. Indeed, I was initially inclined to question the result, but the experimental design and conduct of the experiments seem solid. The over-zealous tone of the presentation detracts from the impact of the study.

We thank reviewer 2 for this advice and have attempted to dial down the tone in our revised manuscript from over-zealous to merely excited by our results.

The authors tend to overstate the poor ITD sensitivity of cochlear implant (CI) users. In the Introduction, for instance, states that "ITD sensitivity of CI patients is poor or completely missing". This might be more-or- less true of pre-lingually deaf CI patients. It is not correct, however, for post-lingually deaf patients, and the authors don't make that distinction here. Figure 2 of Laback et al., 2011, for instance, shows that the median ITD threshold for post-lingual patients is between 100 and 200 microsec and that post-lingual patients with no ITD sensitivity are rare (about 9% in that figure). Thresholds of 100-200 microsec are comparable to those of normal hearing human listeners detecting interaural delays in sound envelopes. Again, in subsection “Early deaf CI rats discriminate ITD as accurately as their normally hearing litter mates”, I think that it is not correct that pre-lingual CI patients "usually" have ITD thresholds "too large to measure", although I would accept "often".

We thank reviewer 2 for this comment and have revised the two sentences in the Introduction accordingly. We are of course happy to tone down “usually” to “often”, or to flag the particular difficulties of pre-lingually deaf patients, who are in fact our main interest group. Nor do we wish to get into an argument about quite how poor the ITD of CI users really is. It is generally accepted that human bilateral CI users struggle with ITDs, and that this difficulty tends to be particularly pronounced in early deaf patients. That provides sufficient motivation for our research.

It is important to note that ITD sensitivity in CI patients declines as pulse rates are increased above about 50 pps, and clinical processors run at >900 pps. The present study used single bilateral pulse pairs for physiology and relatively low (50 pps) pulse rates for behavior. For that reason, the present results in rats cannot be compared with human results obtained with clinical processors, only with those made with laboratory processors.

Reviewer 2 is of course quite correct that pulse rate matters (we are currently conducting follow on studies documenting this in our animal model), and we now have incorporated this point in our revised Discussion. Indeed, there are many additional factors that would need to be taken into account if one wished to formally compare our animal results against those obtained with human patients. But here we are not trying to make a detailed, formal comparison between our animal results and any one particular human study or patient group. Instead, we hope the reviewer will concede that the behavioral performance exhibited by our ND rats is clearly surprisingly good compared with any human patient performance data published so far.

There are as many as 5 experimental groups of rats that are lumped together as neonatally deafened (ND) and normal hearing (NH). I count the following groups:1) Neonatally deafened, trained and tested for behavior2) Neonatally deafened, tested with inferior colliculus (IC) recording3) Raised with normal hearing, trained and tested for behavior using sound stimulation4) Raised with normal hearing, tested with IC recording and sound stimulation5) Raised with normal hearing, deafened as adults (I think), implanted with ICs, and tested with IC recording – These animals are referred as "NH rats", the same as to groups 3 and 4

We thank reviewer 2 for the important hint that the experimental groups of our study were not as clearly defined and named as one might wish. We have remedied this in the revised paper and added a new figure to the revised manuscript (see new Figure 1) which provides a simple overview of the four different experimental groups and the experimental pipeline they followed. Our study does not include any group of acoustically stimulated rats with IC multi-unit recording. Both groups with IC recording were bilaterally supplied with CIs immediately before the start of the IC measurements and thus electrically stimulated during the IC recording to minimize the number of variables. In fact, the only difference between the groups was their hearing experience: neonatally deafened (ND) versus normal hearing until CI implantation (now termed as hearing experienced (HE)). Please see group 3 and 4 in the new Figure 1.

We need to know if groups 1 and 2 are the same rats. If so, what was the history of electrical stimulation prior to IC recording? That is, did they get just 2 25-30-minute sessions a day of stimulation for about 14 days, or did they receive some other electrical stimulation?

The animals undergoing electrophysiology recordings were not the same as those undergoing behavioural testing, but were their littermates. The animals used for physiology had no experience of CI stimulation prior to the recording session. Those used for behaviour were only stimulated during their twice daily training and testing sessions. The new Figure 1 and clarifications in subsection “Varying degrees and types of ITD tuning are pervasive in the neural responses in the IC of ND rats immediately after adult cochlear implantation” of the revised paper will hopefully resolve any confusion about the stimulation history of any group of animals described in the paper.

Were group 5 rats deafened, implanted, and recorded all the same day, or did time elapse between implantation and IC recording? Were those rats deafened and, if so, how?

The animals described as “Group 5” by reviewer 2 correspond to what we now call group HECI-E (see new Figure 1). These animals were normally hearing until bilateral CI implantation, and thus hearing experienced. They were not deafened, and IC recordings began effectively immediately after conclusion of the implantation of CI and recording electrodes. The whole experiment was performed in one day. We added the following information to our revised manuscript (subsection “Varying degrees and types of ITD tuning are pervasive in the neural responses in the IC of ND rats immediately after adult cochlear implantation”). These animals were not chemically deafened. However, during CI implantation, their tympanic membrane was perforated and, the middle ear ossicles removed (which will immediately raise hearing thresholds by 50 dB (Illing and Michler, 2001)), the cochlea was opened above the middle turn, the CI array inserted, and then the middle ear and auditory canal were filled with agar. This would have dramatically impaired any residual hearing. Also, physiological responses were recorded in a sound attenuated chamber in the absence of any acoustic stimulation. We now added this information to the subsection “Multi-unit recording from IC”.

We need a different label to distinguish group 5 from groups 3 and 4. The caption of Figure 3 refers to "hearing experienced" rats and cats, which I think is a useful term. Still, the presentation of Figure 3 in the text refers only to "NH", and it is not obvious that the data in Figure 3 are, I guess, from rats that were raised with normal hearing, then deafened, implanted, and stimulated electrically as young adults.

We thank reviewer 2 for this hint and have now introduced four new abbreviations for the four experimental groups of our study (see new Figure 1) and used them in our revised manuscript and the Figures to be able to distinguish clearly between the different groups. reviewer 2 has correctly understood Figure 3, the rats were normal hearing and therefore hearing experienced until the time of bilateral CI implantation which induced acute deafness. The labels of Figure 4 have been corrected as per the new Figure 1.

Reviewer #3:This paper has significance in the fields of Developmental Biology and Neuroscience and a warrants publication in eLife. The aim of the current study, using behavioral and physiological evidence, was to determine if interaural time difference (ITD) thresholds can be measured in neonatally deafened rats when implanted with cochlear implants in adulthood. In addition, the rats were provided with rigorous training to ITDs. A secondary aim of the study was to debunk the idea of an early "critical period" for learning ITDs. The study found that neonatally deafened rats were able to behaviorally respond and learn to use ITDs with exquisite precision even when no input is provided during an early "critical period" and that this was due to the presentation of consistent, salient ITDs after cochlear implantation.As a basic science question, the work in this study provides an excellent animal model for future behavioral testing in binaural tasks. In addition, the methods and implementation are entirely sound and the work is well-motivated, and falls within the scope of this journal. Clinically, there is a need in the field to understand the impact of bilateral implantation on the developing auditory system and its ability to provide the necessary cues for sound localization. This study provides a good stepping stone for future developmental studies that would like to probe whether better processing strategies, or early intervention with cochlear implants, is the more pressing issue for ascertaining ITD sensitivity. However, there were major logical flaws or gaps in the manuscript. These could easily be fixed with changes to the language in the Introduction and Discussion. These major issues are outlined below:1) The concept of the "critical period" is not well-defined. More of the literature on the relationship between critical periods and ITD sensitivity should be surveyed in order to understand what the authors are referring to in the context of the present study. First, it would be very helpful to the reader to be able to visualize, with a schematic or timeline, how the animals in the current study were, deafened, implanted, and subsequently trained with ITDs, compared with other studies.

We thank reviewer 3 for this suggestion and have added a new Figure 1 for a better overview of our four experimental groups which shows the time and type of treatment for each of the cohorts.

Second, what is this nature of this critical period? Is it a period for development of a skill, a vulnerable period, a period for recovery from deprivation? These examples are taken from Kral 2013 as examples of previously published definitions. I think it might be in the authors' benefit to clearly define what this critical period represents so that the interpretation of the results can be easily understood.

We thank the reviewer for this excellent suggestion. The critical period referred to here is indeed as defined in Kral, (2013). To clarify this definition, we have inserted the following sentence in the Introduction: “In other words, if there are no binaural inputs with useful ITDs during a presumed "critical" developmental period, then the neural circuitry needed for ITD sensitivity is thought to fail to develop, leading to a perhaps permanent deficit (Kral, 2013).”

2) Throughout the Introduction and Discussion there is an implicit suggestion about how deafness at an early age may not necessarily cause a loss of ITD sensitivity, and instead that CIs have limitations which could prevent learning of ITDs. However, a distinction needs to be made on what type of limitations are being debated: is it (a) the inability of CI to provide ITDs or is it the (b) exposure to inconsistent ITDs through bilateral CIs that is preventing learning of those cues? In the manuscript, the authors appear to advocate for the latter point. This distinction is further confused by the sentence in the Introduction, where the sensitivity of CI patients is mentioned, citing papers with a mix of measurements made with both clinical processors and synchronized research processors. Therefore, if the authors are suggesting that poor delivery of ITDs using unsynchronized processors is preventing learning of ITD cues in humans, then an equivalent condition with unsynchronized processors needs to be tested in the rats. This must be done in order to confidentially say that synchronized ITDs through the CI need to be conveyed in order to promote learning of binaural cues.

We agree entirely with reviewer 3 and have tried to remedy the points where our manuscript is not clear. We would like to suggest that the possibility a) put forward by the reviewer is not one of practical relevance. If a patient is fitted with CIs in both ears then the stimuli delivered to the left and right ear respectively will inevitably have some temporal relationship and hence there have to be ITDs. Every form of binaural stimulation must necessarily “provide ITDs”, but whether these ITDs are delivered in a manner that is consistently informative and usable by the patients’ auditory system is of course a very different question. We are therefore ultimately exclusively interested in question b). But fully answering that question was not the objective of this study. Here we merely aim to falsify the critical period hypothesis. Fully mapping out what quality of ITDs need to be provided when in order to achieve good ITD sensitivity through prosthetic hearing will require several follow-on studies with appropriately chosen controls and age cohorts. With our study we have laid the foundation for this important work by developing the first valid behavioural, bilateral CI animal model which clearly, at least under ideal conditions can develop very high levels of ITD sensitivity despite severe deafness during hypothetical critical periods. In the revised manuscript we have tried to be more careful not to overstate the implications of the present set of results, and to stress the importance of future studies designed to contrast the effect of different binaural simulation regimes on eventual ITD sensitivity outcomes.

3) A final consideration, the authors should clearly state the hypotheses of the study. The authors have designed a unique study with two experiments, one behavioral and one physiological, whereby response lateralization and the proportion of ITD sensitive neurons were used as a metric to understand whether ND rats were able to (a) utilize ITDs and (b) possessed the underlying neural structures which reflect the behavioral data. It is imperative for the authors to state a set of hypotheses in relation to these two experiments and subsequently describe to the reader how they were carried out and measured. This means that, somewhere in the Introduction, there should be a justification for measuring response lateralization and proportion of ITD sensitive neurons/SNR for the behavioral and physiological experiments, respectively.

We thank reviewer 3 for this valuable suggestion. We have thoroughly revised the Introduction to make it clear that the principal aim of our study is simply to test the critical period hypothesis through our behavioral work. In the event, our results falsify that hypothesis. The electrophysiology data are presented as “corroborating and illustrative” data, showing that midbrains of ND rats have a high amount of ITD sensitivity, which is hardly surprising given how well these animals can learn to lateralize ITDs. But the electrophysiology was not set up as a formal hypothesis test.

To incorporate reviewer 3’s suggestion, we reworked the Introduction, and the penultimate paragraph of the Introduction now reads:

"In summary, the critical period hypothesis is widely considered a likely explanation for the poor ITD sensitivity reported in binaural CI patients, but it has not been subjected to attempts of direct experimental falsification. Therefore, our objective here was to examine experimentally the contrary hypothesis, namely that very good ITD sensitivity can be induced even after periods of severe hearing loss throughout infancy, provided that the stimulation used in early rehabilitation is optimized for ITD encoding. Given the great difficulties described in examining this hypothesis in a clinical setting, we sought to test this hypothesis by developing an animal model which is amenable to psychoacoustic behavior testing."

Introduction:Needs a thorough literature review on the auditory critical period in relation to ITD sensitivity. The authors should also discuss why there hasn't been a rigorous study to understand critical periods and ITD sensitivity.

We have tried hard to accommodate the reviewer’s request, and extended our discussion of previous work on ITD critical periods both in the Introduction and in the Discussion section. We have also tried to highlight some of the difficulties that would be involved in studying critical periods in a clinical setting, but we aren’t sure whether that meets the reviewer’s demand that we “discuss why there hasn't been a rigorous study to understand critical periods and ITD sensitivity”. Ultimately, the reasons why the research community hasn’t yet mapped out ITD critical periods are not knowable, but we have tried to say something relevant to this topic.

On the topic "critical period for the development of ITD sensitivity" Seidl and Grothe, 2002 showed in gerbils that this sensitivity matures only after the onset of hearing and that this development can be dramatically disturbed by altered acoustic input such as omnidirectional white noise between P10 and P25 with an ITD tuning that resembles an immature, juvenile system. If the normal hearing animals were exposed to noise in the adult, no changes were observed. The authors concluded that the development of ITD tuning requires normal acoustic experience in the early stages of development to possibly develop the inhibitory inputs of the ITD system. However, the study leaves open whether an early deafened and thus inexperienced system can still mature through sensory activation at a later stage, including the maturation of inhibitory inputs. Important evidence for such an adaptability of the adult auditory system was found in one of our earlier studies on neonatally deafened rats that were supplied with CIs as young adults (Rosskothen-Kuhl et al., 2018, ). Here, we show that the initial activation of the auditory system by electrical stimulation significantly upregulates inhibitory markers such as GAD65 and GAD67 in the auditory midbrain and thus, possibly also in an adult brain, important input-induced maturation processes can take place that may allow the development or fine tuning of ITD sensitivity.

Need to make distinction between early-deafened vs. late-deafened. This will help to understand the relation between human and rats.

This also relates to criticisms made by reviewer 2. In the thoroughly revised Introduction we have greatly expanded the description of the differences between these two patient groups.

Needs more information on training and plasticity in order to build up the question. While prior studies have shown that early deafness does impact ITD sensitivity, they do not suggest that sensitivity could not be relearned. Please describe evidence (if any) that supports the notion that lack of exposure to binaural cues in adulthood, rather than a total loss of binaural function early in life, limits sensitivity to binaural cues for early-deafened populations.As stated earlier, a distinction should be made between synchronized delivery of binaural cues and unsynchronized bilateral implantation and which one of these is the contributing factor to improper learning or acquisition of ITDs in adulthood.

The extent to which it may be possible to relearn ITD sensitivity has not been explored in great detail, but we do know that simply clocking up more and more bilateral CI hearing experience with conventional processors does not lead to high levels of ITD sensitivity, as has been shown by studies such as Gordon et al., (2014). These have been incorporated in the extended Discussion.

Subsection “Varying degrees and types of ITD tuning are pervasive in the neural responses in the IC of ND rats immediately after adult cochlear implantation”: The interpretation of various shapes of tuning curves aren't always known to the average reader. It would be helpful, either here or in the data analysis section, to describe what each of the curve types described about the neurons that are being recorded from.

Interpreting tuning curve shapes is perhaps more an art than a science. Comparing it to reading tea leaves would perhaps be unfair, as there have been rigorous attempts to think about, for example what tuning curve shapes one might expect to see under various constraints and assumptions about optimal neural coding (eg. Harper and McAlpine, 2004) but the issues are complex and not all that relevant to the issues examined in this study. We merely wanted to point out that the tuning curves we observed are somewhat similar to those described by previous authors working on similar preparations (e.g. Smith and Delgutte, 2007). Please see subsection “Varying degrees and types of ITD tuning are pervasive in the neural responses in the IC of ND rats immediately after adult cochlear implantation”. We have tried to be clearer on our description of the tuning curve shapes and reworked the subsection “Data analysis” and hope that meets the reviewer’s requirements.

Discussion: Please expand on the importance of ITD processing in rats and what "generally poor" means in the context of the current study.

We expanded on the importance on ITD processing for rats and have clarified this sentence (Discussion).

Discussion: The logic here is a little hard to follow. The authors state that only 48% of IC neurons were found to be ITD sensitivity in ND subjects, conversely in the current study, 91% of IC neurons were ITD sensitive. The claim appears to be that because the proportion of ITD sensitive neurons was greater in the present study, that this is a hallmark of relearning of ITDs in the animal. However, nowhere in the introduction or the hypotheses of the paper was this addressed. The hypotheses of paper need to be clearly stated such that the reader can understand what measures from the (a) behavioral and (b) physiological data constitute "good sensitivity to ITDs".

We thank reviewer 3 for pointing out this misunderstanding here. As shown for group 3 in our new Figure 1, the IC recordings were made in neonatally deafened rats bilateral implanted with CIs as young adults. These rats had no previous hearing experience with electrical CI stimulation before IC recording. The 91% sensitive IC units were thus obtained without “relearning of ITDs” in these animals and our intention was to point out that in the optimal stimulation conditions (microsecond precise stimulation) used in this study even in a naive auditory system innate ITD sensitivity exists.

The objective of this study has been added at the end of the Introduction. Additionally, we have added some clarity to these sentences to improve the logic flow (Discussion).

Discussion: This sentence, "It is important to remember that…congenital cats" is very long and a little difficult to follow. I think it would be helpful if the authors previously described what characterized good vs. poor tuning. It sounds like the authors are trying to make a connection between the behavioral data and the neural data, however, this is unclear.

We have broken down this sentence and rewrote this paragraph in order to make the argumentation more clear (Discussion).

Discussion: There is a reference to "critical periods" in each of these animal models, however, the statements made are rather vague. In each the ferret, mouse, and gerbil models, there is a reference to certain synapses and their relationship to binaural circuity prior to the onset of hearing (i.e. the "critical period"). However, the authors do not explain what happens to the circuitry, and what is required to occur (either before or after the so-called "critical period") in order to overcome any deficits in ITD processing. In addition, the authors have not explicitly stated what makes the rat model in present study unique for understanding the implications of the hypothesized "critical period."

We thank the reviewer for pointing this out, and we have revised the Discussion and added the necessary clarifications.

Our animal model is unique because it allows for the first time to test ITD sensitivity behaviorally with a high degree of accuracy and sensitivity in an animal model in which hearing experience as well as stimulation and training or rehabilitation parameters can be freely manipulated and controlled. This has enabled us to show, in a first instance, that despite the lack of sensory input during a presumed critical period, the neonatally deaf auditory system can develop the ability to lateralize ITDs with high accuracy. Follow on studies will enable us and others to map out the conditions that determine eventual binaural auditory performance outcomes in detail.

Discussion: Authors should expand on why inhibition is important.

We have added additional information from Pecka et al., (2008) showing the importance of inhibition for ITD processing (Discussion).

[Editors' note: further revisions were suggested prior to acceptance, as described below.]

Summary:The reviewers agree that the authors have made substantial improvements, especially with toning back the maladaptive plasticity hypothesis. However, there are two major remaining issues.First, the writing of the Introduction still needs significant work, with more specific and crisper language, especially when referring to literature or ideas, such as when discussing the contribution of the critical period and related hypotheses as well as the role of training, as noted by reviewer 3. The authors also need to clarify interpretation of the rats' ability to use ITDs with exquisite precision as due to (1) provision of salient ITDs (independent of missed critical period) or (2) rigorous training and synchronized stimulation allowed rats to learn to use ITDs even though the critical period was missed.

We thank reviewer 3 for this feedback. With regard to the last point mentioned above, we would like to point out that it does, of course, to an extent, depend on how one defines a critical period. When we wrote the paper, we had the example of scholars of binocular vision in mind, who, as a general rule, think of critical periods “strong”, in the sense that almost no amount of training after closure of the critical period can fully compensate for the deficits incurred by the lack of appropriate input during the critical period. Consider that essentially no adult amblyopic patients ever achieve normal stereopsis or normal visual acuity in the affected eye, irrespective of the quality of the visual stimulation and rehabilitative training they receive after the critical period has closed. If the situation in CI ITD sensitivity was the same as that in amblyopia, then no amount of training could restore near normal ITD sensitivity in adulthood. Given that our adult implanted CI rats achieved ITD sensitivity that was no worse than that of normally hearing litter mates can only mean that there is no “strong” critical period for ITD sensitivity development in young rats. Of course, high quality stimulation and training too might play a role in the case of a “weak” critical period, but firstly, the small amounts of training needed for our behavioral ND rat cohort indicates that a critical period, if it exists at all, would have to be pretty weak indeed, and secondly, any “weak” critical period could only ever have weak explanatory power when we are trying to understand the causes of poor binaural outcomes for current early deaf CI patients. We therefore consider the weak critical period case scientifically and clinically uninteresting. We have substantially rewritten the Introduction to make the analogy to amblyopia explicit and emphasize that the aim of our study is to test the “strong critical period hypothesis”. That resolves the potential ambiguity in how our results should be interpreted which the reviewer had pointed out here.

There is also a new concern raised by reviewer 2 about the HE-CI group which the revision indicates were not chemically deafened before implantation. Specifically, there is a potential for electrophonic responses (e.g. recent work by Kral's group (Sato et al., 2016). The acoustic frequency corresponding to the parameters used in this paper would be approximately the period of the pulse, in this case 1/0.000164 or ~6000 Hz. It is not clear what tonotopic areas were recorded in the study and how electrophonic responses would affect the result. Even though the HE-CI group was given a conductive loss, this would not attenuate the signal as completely as chemical deafening, and inner/outer hair cells are likely to remain intact. and permit indirect electrophonic stimulation of inner hair cells via the outer hair cells, as well as direct electrical stimulation of inner hair cells. After discussion, all of the reviewers agreed that this is a substantial experimental confound, and also that this group does not add value to the study. Therefore, we recommend to remove this group.

We have followed the suggestion and removed data from the HE-CI group from this study and adapted Figure 1, Figure 3, and Figure 4 accordingly.